# A novel SH2 recognition mechanism recruits Spt6 to the doubly phosphorylated RNA polymerase II linker at sites of transcription

Matthew A Sdano[1], James M Fulcher[1], Sowmiya Palani[2,3], Mahesh B Chandrasekharan[2,3], Timothy J Parnell[2,3,4], Frank G Whitby[1], Tim Formosa[1]*, Christopher P Hill[1]*

[1]Department of Biochemistry, University of Utah School of Medicine, Salt Lake City, United States; [2]Department of Radiation Oncology, University of Utah School of Medicine, Salt Lake City, United States; [3]Huntsman Cancer Institute, University of Utah School of Medicine, Salt Lake City, United States; [4]Department of Oncological Sciences, University of Utah School of Medicine, Salt Lake City, United States

**Abstract** We determined that the tandem SH2 domain of *S. cerevisiae* Spt6 binds the linker region of the RNA polymerase II subunit Rpb1 rather than the expected sites in its heptad repeat domain. The 4 nM binding affinity requires phosphorylation at Rpb1 S1493 and either T1471 or Y1473. Crystal structures showed that pT1471 binds the canonical SH2 pY site while pS1493 binds an unanticipated pocket 70 Å distant. Remarkably, the pT1471 phosphate occupies the phosphate-binding site of a canonical pY complex, while Y1473 occupies the position of a canonical pY side chain, with the combination of pT and Y mimicking a pY moiety. Biochemical data and modeling indicate that pY1473 can form an equivalent interaction, and we find that pT1471/pS1493 and pY1473/pS1493 combinations occur in vivo. ChIP-seq and genetic analyses demonstrate the importance of these interactions for recruitment of Spt6 to sites of transcription and for the maintenance of repressive chromatin.
DOI: https://doi.org/10.7554/eLife.28723.001

*For correspondence:
tim@biochem.utah.edu (TF);
chris@biochem.utah.edu (CPH)

## Introduction

RNAPII is the twelve-subunit complex that transcribes mRNAs and many noncoding RNAs in eukaryotes (*Kornberg, 1999*). The largest subunit, Rpb1, contains a flexible C-terminal domain (CTD) comprising multiple repeats of the heptad consensus sequence YSPTSPS, with about 26 repeats in *S. cerevisiae* and 52 in human (*Eick and Geyer, 2013*). Phosphorylation of CTD residues is thought to recruit multiple co-factors, including the essential and conserved transcription factor Spt6, which directly binds to and co-localizes with RNAPII at sites of transcription (*Andrulis et al., 2000*; *Kaplan et al., 2005*; *Kaplan et al., 2000*; *Kim et al., 2004*; *Mayer et al., 2010*; *Perales et al., 2013*; *Yoh et al., 2007*; *Yoh et al., 2008*). Spt6 has been implicated in multiple steps of gene expression including transcription, mRNA processing and export, histone post-translational modification, and nucleosome positioning (*Duina, 2011*). A well-established Spt6 activity that is thought to depend upon its association with RNAPII is the reassembly of nucleosomes in the wake of elongating RNAPII, which is required to maintain chromatin in the repressive state that prevents aberrant transcription initiation (*Adkins and Tyler, 2006*; *DeGennaro et al., 2013*; *Hainer et al., 2011*; *Ivanovska et al., 2011*; *Kaplan et al., 2003*; *Thebault et al., 2011*).

The 1451 residues of Spt6 comprise three structural regions (*Close et al., 2011*). The N-terminal ~300 residues are highly acidic, predicted to be disordered, and are necessary for binding both nucleosomes and the transcription factor Spn1/IWS1 (*Diebold et al., 2010a*; *McDonald et al., 2010*). The core of Spt6 (residues 298–1248) contains several structural motifs associated with binding to DNA or proteins, and has overall similarity to the prokaryotic transcription factor Tex (*Johnson et al., 2008*). The C-terminal region (residues 1250–1440) comprises a tandem SH2 (tSH2) domain that is tethered to the core by a flexible 21-residue helix and contains both of the two known SH2 structural motifs in yeast (*Close et al., 2011*; *Diebold et al., 2010b*; *Sun et al., 2010*). These two SH2 motifs pack against each other to form a single structural unit. The N-terminal module (nSH2) resembles a canonical SH2 domain while the C-terminal module (cSH2) is vestigial and does not conserve the residues that are normally associated with binding to pY-containing peptides that comprise the vast majority of known SH2 domain ligands (*Liu et al., 2006*).

The tSH2 domain is important for Spt6 function. In yeast, deletion of the tSH2 domain causes slow growth and phenotypes attributed to defects in transcription elongation or maintenance of chromatin status (*Diebold et al., 2010b*; *Hartzog et al., 1998*; *McCullough et al., 2015*; *Sun et al., 2010*). In mammals, the tSH2 domain is important for mRNA processing and export, and for class switch recombination (*Begum et al., 2012*; *Okazaki et al., 2011*; *Yoh et al., 2007*). Furthermore, deletion of the tSH2 domain reduces occupancy of Spt6 throughout transcribed genes (*Burugula et al., 2014*; *Mayer et al., 2010*). Although the tSH2 domain was initially reported to bind the serine-2 phosphorylated Rpb1 CTD (*Yoh et al., 2007*; *Yoh et al., 2008*), the conservation of the nSH2 domain implies that it binds a pY residue, and the presence of this modification in the CTD heptad repeats provides an attractive model for recruitment of Spt6 to RNAPII (*Mayer et al., 2012*). This widely accepted model is supported by a number of studies that have reported binding of the tSH2 domain to phosphorylated RNAPII CTD peptides (*Close et al., 2011*; *Liu et al., 2011*; *Mayer et al., 2012*; *Sun et al., 2010*).

Despite its wide acceptance, a number of observations caused us to question the prevailing model: (1) Affinity of the tSH2 domain for phosphorylated CTD-derived peptides is weak under physiological salt conditions, (2) binding displayed little preference for different phosphorylation states, and (3) mutations expected to disrupt pY interaction had only modest effects on binding and no apparent physiological consequences (*Close et al., 2011*). We therefore took an unbiased approach to map the tSH2 domain binding site within RNAPII and, contrary to the earlier assumptions, found that Spt6 tSH2 directly binds the flexible, 86 residue Rpb1 linker (residues 1456–1541) that tethers the heptad CTD repeats to the RNAPII core. Biochemical and structural studies demonstrate that optimal binding requires phosphorylation of the Rpb1 linker on S1493 as well as either T1471 or Y1473. The interaction with pS1493 occurs through a novel, unanticipated site, while phosphorylated T1471/Y1473 binds to the canonical SH2-pY-binding site. The pT1471 interaction displays remarkable molecular mimicry with canonical pY complexes, and suggests both an evolutionary pathway to the extensive family of SH2-pY complexes of metazoans and the possibility of novel SH2 signaling paradigms. Mass spectrometry showed that phosphorylation can occur in vivo simultaneously on T1471 and S1493 or on Y1473 and S1493. ChIP-seq revealed that disruption of the Spt6-Rpb1 interface reduced Spt6 occupancy of transcription units, especially at highly transcribed genes. Genetic analysis demonstrated the functional importance of the Spt6-RNAPII interaction for the maintenance of repressive chromatin. These studies therefore uncover the mechanism through which Spt6 is recruited to transcription sites, and reveal an unexpectedly complex phosphorylation-dependent switch for this recruitment that has strong implications for other SH2-mediated processes.

## Results

### The Rpb1 linker, but not the CTD region, is required for Spt6 tSH2 binding

To test the importance of the heptad repeats of Rpb1 for binding the Spt6 tSH2 domain, we inserted PreScission protease sites into Rpb1 to facilitate removal of the CTD (ΔCTD) or linker and CTD (Δlinker) during purification from yeast cells (*Figure 1A and B*). Purified RNAPII proteins were transferred to a membrane and probed with purified recombinant GST-Spt6$^{1223\text{-}1451}$ (GST-tSH2) in a

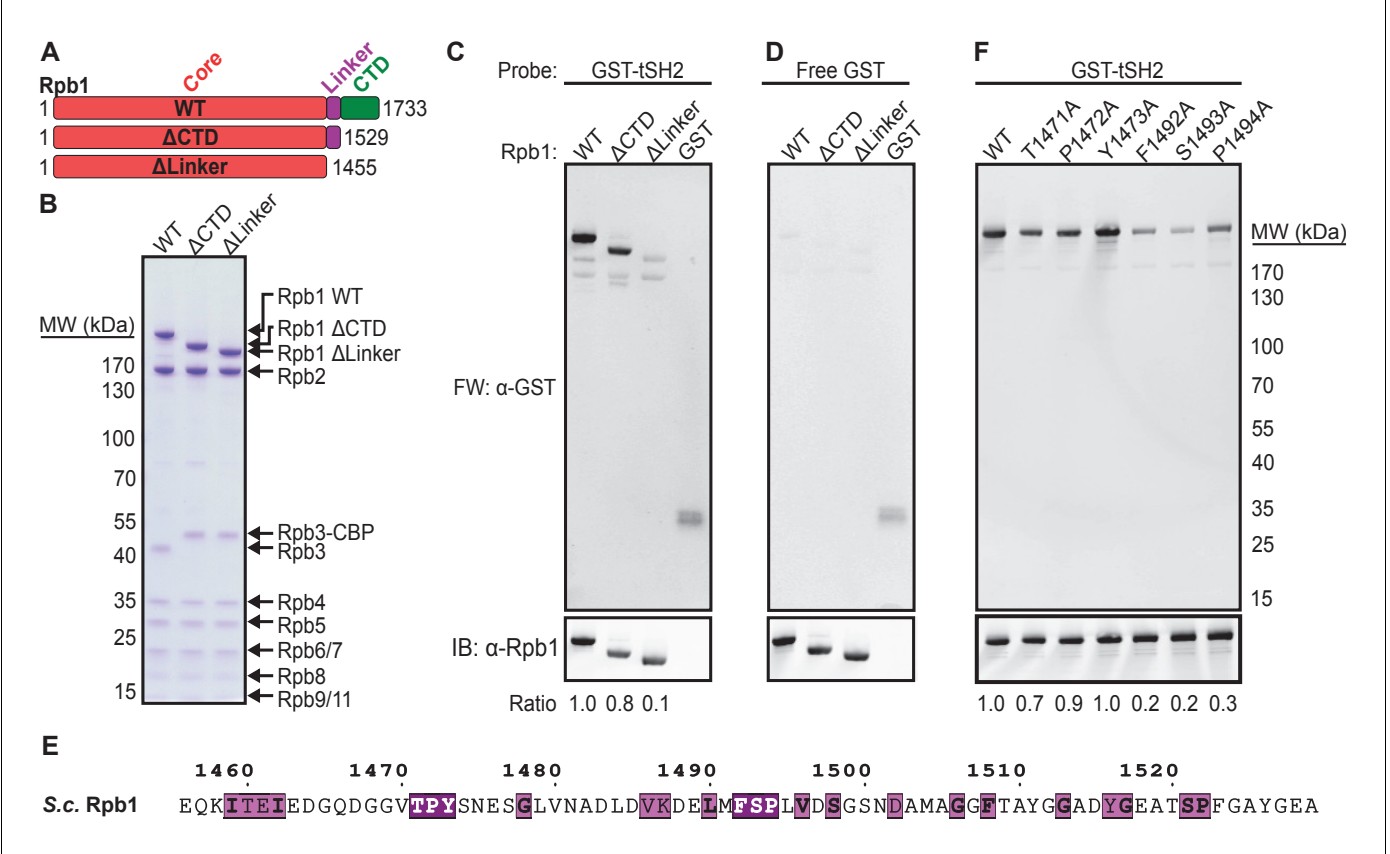

**Figure 1.** Conserved residues in the Rpb1 linker are necessary for binding Spt6 tSH2. (**A**) Full-length Rpb1 (WT) and deletion constructs (ΔCTD and ΔLinker) tested for binding. (**B**) Coomassie-stained SDS-PAGE of purified RNAPII with the Rpb1 variants depicted in panel A. Rpb3 from ΔCTD and ΔLinker strains retains a TAP calmodulin-binding peptide (CBP) tag not found in wild type. (**C**) Proteins in (**B**) were probed with GST-Spt6 tSH2 (Far Western; FW) or an antibody against the Rbp1 N-terminus (y-80; Santa Cruz Biotechnology). The ratio of signal for the far western over α-Rpb1 normalized to WT is indicated. (**D**) As in (**C**), except probed with free GST. (**E**) Sequence of the *Saccharomyces cerevisiae* (*S.c.*) Rpb1 linker colored by conservation across *Saccharomyces cerevisiae, Schizosaccharomyces pombe, Caenorhabditis elegans, Drosophila melanogaster, Danio rerio*, and *Homo sapiens* aligned using T-Coffee (RRID:SCR_011818; *Notredame et al., 2000*) and visualized using ESPript (RRID:SCR_006587; *Robert and Gouet, 2014*). White on purple represents strictly conserved residues, black on magenta is a global similarity score ≥0.7. (**F**) Far western blot (FW) measuring binding of Spt6 tSH2 to point mutants in the Rpb1 linker. Ratios were determined as in (**C**) except that the antibody 8WG16 against the Rpb1 CTD was used (Covance).

DOI: https://doi.org/10.7554/eLife.28723.002

far western assay. Consistent with previous observations (*Yoh et al., 2007*), GST-tSH2 bound full length Rpb1 (residues 1–1733) specifically (*Figure 1C and D*). Surprisingly, Rpb1 ΔCTD (residues 1–1529) bound equally well, indicating that the heptad repeats are not necessary for this interaction. Deleting an additional 74 residues to remove the linker that tethers the CTD to the RNAPII core abolished binding (*Figure 1C*), indicating that this region contains the primary binding site.

Two clusters of highly conserved residues within the linker were considered as candidates for binding sites (TPY$^{1471-1473}$, and FSP$^{1492-1494}$; *Figure 1E*). Mutating individual residues in the first cluster had little effect on binding in the far western assay, whereas mutating the second cluster caused a substantial reduction (5-fold for Rpb1$^{S1493A}$; *Figure 1F*). Conserved residues in the Rpb1 linker therefore contribute significantly to Spt6 tSH2 binding in this assay while the CTD heptad repeats do not.

## Phosphorylation of Rpb1 S1493 promotes tSH2 binding

To determine if the Rpb1 linker is sufficient for Spt6 tSH2 binding, we used fluorescence polarization to quantify binding to a fluorescein-labeled peptide spanning the FSP$^{1492-1494}$ residues implicated in

the previous section (Fl-Rpb1$^{1476-1511}$). Spt6$^{1247-1451}$ (tSH2) bound the peptide in solution with a $K_D$ of ~43 μM (*Figure 2A*), which is unlikely to explain the binding observed in the far western assay. Direct binding of Spt6 tSH2 to fluorescein itself at concentrations above ~15 μM prevented accurate testing of interactions weaker than 2 μM (data not shown). Notably, this effect may have influenced previously published data measuring binding of tSH2 to fluorescein-labeled CTD peptides. Given the reported importance of RNAPII phosphorylation in Spt6 binding (*Yoh et al., 2007*; *Yoh et al., 2008*) and the strong effect noted with mutation of S1493, we next tested the binding to Fl-Rpb1$^{1476-1511}$ with S1493 phosphorylated (Fl-Rpb1$^{1476-1500\ pS1493}$). Phosphorylation increased the affinity ~500 fold, to a $K_D$ of 90 nM (*Figure 2A*), which is approximately three orders of magnitude tighter than reported for monophosphorylated CTD heptad-derived peptides under the same buffer conditions (*Close et al., 2011*). We conclude that Spt6 tSH2 has high affinity for a peptide centered on S1493 of Rpb1, that phosphorylation of this residue is an important binding element, and that

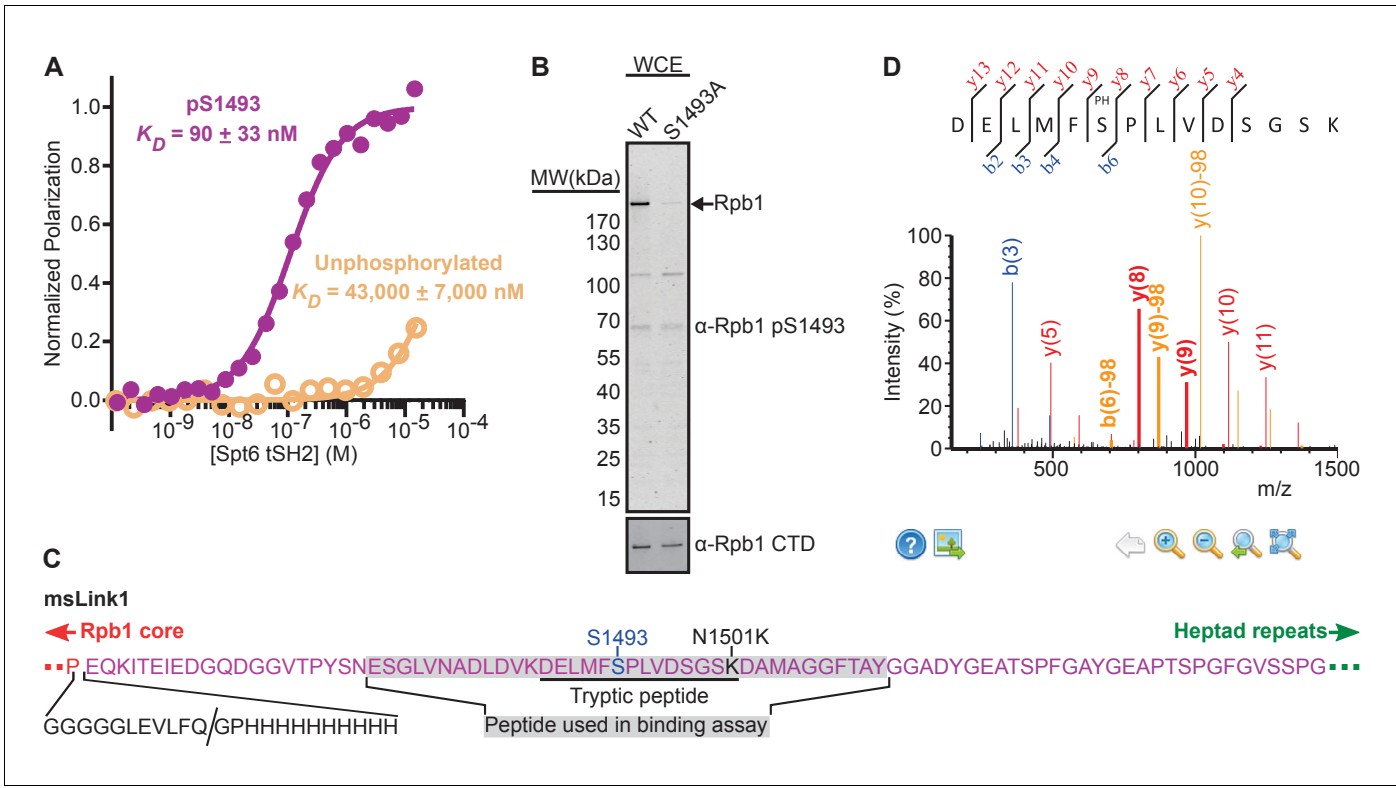

**Figure 2.** Phosphorylation of Rpb1 linker S1493 enhances binding and occurs in vivo. (A) Spt6 tSH2 was titrated into Fl-Rpb1$^{1476-1511}$ peptides with or without S1493 phosphorylated, yielding the indicated changes in polarization. Average binding affinities were calculated from at least 3 independent experiments, with the estimated standard deviations indicated. (B) Whole cell extracts from strains with normal (WT) or mutant (S1493A) Rpb1 were tested by western blotting using antibodies against an Rpb1 pS1493 peptide or the Rpb1 CTD. (C) Sequence of the msLink1 linker. A five-glycine linker, PreScission protease site, and 10x-histidine tag were inserted between P1455 (the last ordered residue in most RNAPII crystal structures; red) and E1456 (beginning of the Rpb1 linker; magenta). The PreScission cut site is indicated with a slash. N1501 was mutated to a lysine (black) to allow generation of a tryptic peptide (underlined) containing S1493 (blue). The peptide used in (A) is indicated with a gray background. (D) Mass spectrum of the peptide underlined in (C), indicating phosphorylation on S1493, after purification of msLink1 from yeast. Peaks representing b ions, blue; y ions, red; neutral loss of 98 Da (phosphate), orange. Peaks above 30% intensity are labeled. Peaks flanking the phosphorylation site are labeled and bold. See also *Figure 2—figure supplement 1* and *Figure 2—source data 1*.

DOI: https://doi.org/10.7554/eLife.28723.003

The following source data and figure supplement are available for figure 2:

**Source data 1.** Binding of fluorescently labeled Rpb1 linker peptides to Spt6 tSH2 wild type protein, related to *Figure 2A*.
DOI: https://doi.org/10.7554/eLife.28723.005

**Figure supplement 1.** The antibody recognizing S1493 phosphorylated Rpb1 does not recognize unphosphorylated Rpb1 peptides.
DOI: https://doi.org/10.7554/eLife.28723.004

at least some of the Rpb1 obtained from yeast that displayed binding to Spt6 tSH2 is likely to be phosphorylated at this site.

Rpb1 S1493 phosphorylation has not been reported previously. We therefore raised an antibody against a phosphorylated linker peptide to ask if this modification occurs in vivo. The specificity of the antibody for the phosphorylated peptide versus the unphosphorylated peptide and a peptide with S1493 replaced with alanine was validated using dot blots with synthesized peptides (*Figure 2— figure supplement 1*). We then tested western blots of yeast whole cell extracts resolved by SDS-PAGE (*Figure 2B*). Several bands were observed, with the predominant band migrating at the same molecular weight as Rpb1. The intensity of this band was greatly diminished when S1493 was mutated to an alanine, confirming that the antibody preferentially recognizes the pS1493 form of Rpb1, and that this modification occurs in vivo.

We next used mass spectrometry to identify phosphorylated tryptic peptides from the Rpb1 linker purified from yeast cells. Rpb1 was modified to allow release and purification of the linker-CTD region from purified RNAPII using PreScission protease, and also to generate suitable tryptic peptides containing the target region (*Figure 2C*). A strain expressing this modified Rpb1 (Rpb1-msLink1) expressed from a plasmid functionally complemented an *RPB1* deletion under a variety of conditions (normal growth at 30°C, 38°C, no Spt$^-$ phenotype, and no sensitivity to hydroxyurea; data not shown). Following purification of msLink1, the engineered tryptic peptide containing S1493, either unphosphorylated or phosphorylated, was readily detected by mass spectrometry, confirming the presence of this modification in vivo (*Figure 2D*). We conclude that some fraction of Rpb1 S1493 is phosphorylated in yeast cells, and that this modification promotes binding of the Spt6 tSH2 domain to Rpb1.

## Structure of an Spt6 tSH2-Rpb1 linker complex

To understand how Spt6 tSH2 preferentially recognizes the Rpb1 linker peptide containing pS1493, we determined a crystal structure of Spt6 tSH2 in complex with Rpb1$^{1476-1500\ pS1493}$. A molecular replacement solution was obtained using the coordinates of unbound yeast Spt6 tSH2 (PDB: 3PSJ [*Close et al., 2011*]) and refined against 2.2 Å resolution data to $R_{work}/R_{free}$ values of 17.7%/24.3% (*Table 1* and *Figure 3*). The crystals contained one complex in the asymmetric unit. Spt6 residues T1250-R1451 and Rpb1 residues L1479-S1498 were visible in the electron density map. Binding to the Rpb1 linker did not induce notable changes in the Spt6 tSH2 conformation, which superimposes on unbound tSH2 (PDB ID: 3PSJ) with an RMSD of 0.52 Å over 189 pairs of Cα atoms.

The structure shows the peptide draping over the tSH2 domain in an extended conformation that buries 1,194 Å$^2$ of solvent-accessible surface area with both hydrophobic and polar contacts contributing to the interaction (*Figure 3*). All of the contacts involve the cSH2 module, and are distant from both the vestigial pY-binding site in the cSH2 module and the canonical pY-binding site of the nSH2 module (*Figure 3A*). The pS1493-binding pocket is formed primarily by Spt6 residues K1355, K1435, and Y1381, with the phosphate moiety forming hydrogen bonding interactions with all three of these side chains (*Figure 3B*). K1435 and Y1381 are strictly conserved among Spt6 homologs, which suggests that the binding geometry is conserved throughout eukaryotes (*Figure 3C*).

S1498 appears to mark the C-terminal boundary of the ligand as it is the last residue observed in this structure as well as in one obtained with a more extended Rpb1 peptide (Rpb1$^{1484-1511\ pS1493}$; data not shown). Rpb1 residues L1479 to F1492 lie against the tSH2 domain in an extended conformation to bury a total of 923 Å$^2$ of solvent-accessible surface area and eight hydrogen bonding interactions. In particular, the side chains of F1492, L1490, and L1484 are buried in a deep hydrophobic groove on the surface of Spt6 tSH2 (*Figure 3B and D*). The structure therefore reveals extensive interactions that explain the high affinity, sequence specificity, and phosphorylation-dependence of Spt6 tSH2 binding to the Rpb1 linker.

## Mutations in the Spt6 tSH2-Rpb1 linker interface disrupt binding in vitro

To validate the crystallographic interface biochemically, we measured binding affinities of purified tSH2 mutants for Rpb1$^{1476-1511\ pS1493}$ using fluorescence polarization (*Figure 3E*). Mutating either one (Spt6$^{K1435A}$) or both (Spt6$^{K1355A,K1435A}$) of the primary phosphate contacts in the binding pocket reduced the affinity for Fl-Rpb1$^{1476-1511\ pS1493}$ approximately 100-fold ($K_D$ 8.3 μM and 25 μM

**Table 1.** Crystallographic Data Collection and Refinement

| Data collection | tSH2-Rpb1$^{pS1493}$ | tSH2-Rpb1$^{pT1471,pS1493}$ |
|---|---|---|
| Beamline | Home source | Home source |
| Wavelength | 1.54178 | 1.54178 |
| Space group | C222$_1$ | C222$_1$ |
| Unit cell dimensions: | | |
| a, b, c (Å) | 43.01, 103.54, 115.49 | 42.45, 102.10, 115.74 |
| Resolution (Å) | 40.00–2.20 (2.28–2.20) | 40.00–1.80 (1.86–1.80) |
| $I/\sigma_I$ | 25 (2) | 8 (1) |
| CC$_{1/2}$ | 0.977 | 0.641 |
| Completeness (%) | 89.81 (60) | 99.67 (96) |
| Redundancy | 24.7 (17.0) | 7.9 (5.2) |
| $R_{meas}$ | 0.056 (0.458) | 0.099 (1.238) |
| $R_{pim}$ | 0.011 (0.106) | 0.035 (0.518) |
| **Refinement** | | |
| Resolution (Å) | 27.64–2.20 (2.28–2.20) | 32.45–1.80 (1.85–1.80) |
| Number of reflections | 12,178 | 23,722 |
| $R_{work}/R_{free}$ (%) | 17.7/24.3 | 16.2/19.8 |
| Number of protein atoms | 1859 | 1954 |
| Number of solvent atoms | 83 | 171 |
| RMSD bond lengths (Å)/angles (°) | 0.007/0.878 | 0.009/1.013 |
| ϕ/ψ most favored/allowed (%) | 96.33/3.67 | 97.75/1.80 |

Values in parentheses refer to the high-resolution shell. Refinement statistics were determined by PHENIX and MolProbity.

DOI: https://doi.org/10.7554/eLife.28723.008

respectively). In contrast, an Spt6$^{R1282H}$ mutation that should inactivate the canonical pY-binding pocket in the nSH2 module had no effect on binding affinity. These results validate the crystallographic interface and demonstrate the importance of this novel pS-binding pocket for supporting the interaction in vitro.

## Phosphorylation of Rpb1 T1471 or Y1473 enhance Spt6 binding

The N-terminal residues of the Rpb1 peptide in our structure extend toward the putative pY-binding pocket of the canonical nSH2 domain. This prompted us to ask whether the nearest phosphorylatable residues upstream might bind this pocket. Y1473 and T1471 are conserved in an alignment of Rpb1 sequences (*Figure 1E*) and would be well positioned to reach the canonical pY-binding pocket of the nSH2 domain. Notably, the msLink1 spectra indicated that T1471 phosphorylation occurs in yeast (*Figure 4A*, *Figure 4—figure supplement 1*), although simultaneous modification with S1493 could not be assessed due to a trypsin cleavage site between these residues. We therefore designed Rpb1-msLink2 to yield a tryptic peptide that contains both residues, and verified that this variant functionally complemented an *RPB1* deletion in yeast. Mass spectrometry of purified msLink2 identified peptides with either T1471 or S1493 phosphorylated as well as peptides with both residues phosphorylated (*Figure 4A*, *Figure 4—figure supplement 1*). Notably, we also observed peptides with Y1473 phosphorylated by itself or in combination with S1493 in the msLink2 dataset, indicating that this residue is also modified in vivo. Collectively, these data indicate that T1471 and Y1473 are phosphorylated in yeast cells and that each individually can be phosphorylated coincident with S1493. Peptides were not observed with both T1471 and Y1473 phosphorylated, although we do not have independent verification that those peptides would be detectable under the mass spectrometry conditions used.

To determine if pT1471 or pY1473 affect tSH2 binding, we used fluorescence polarization with fluorescently labeled, N-terminally extended Rpb1 peptides that also include all of the C-terminal

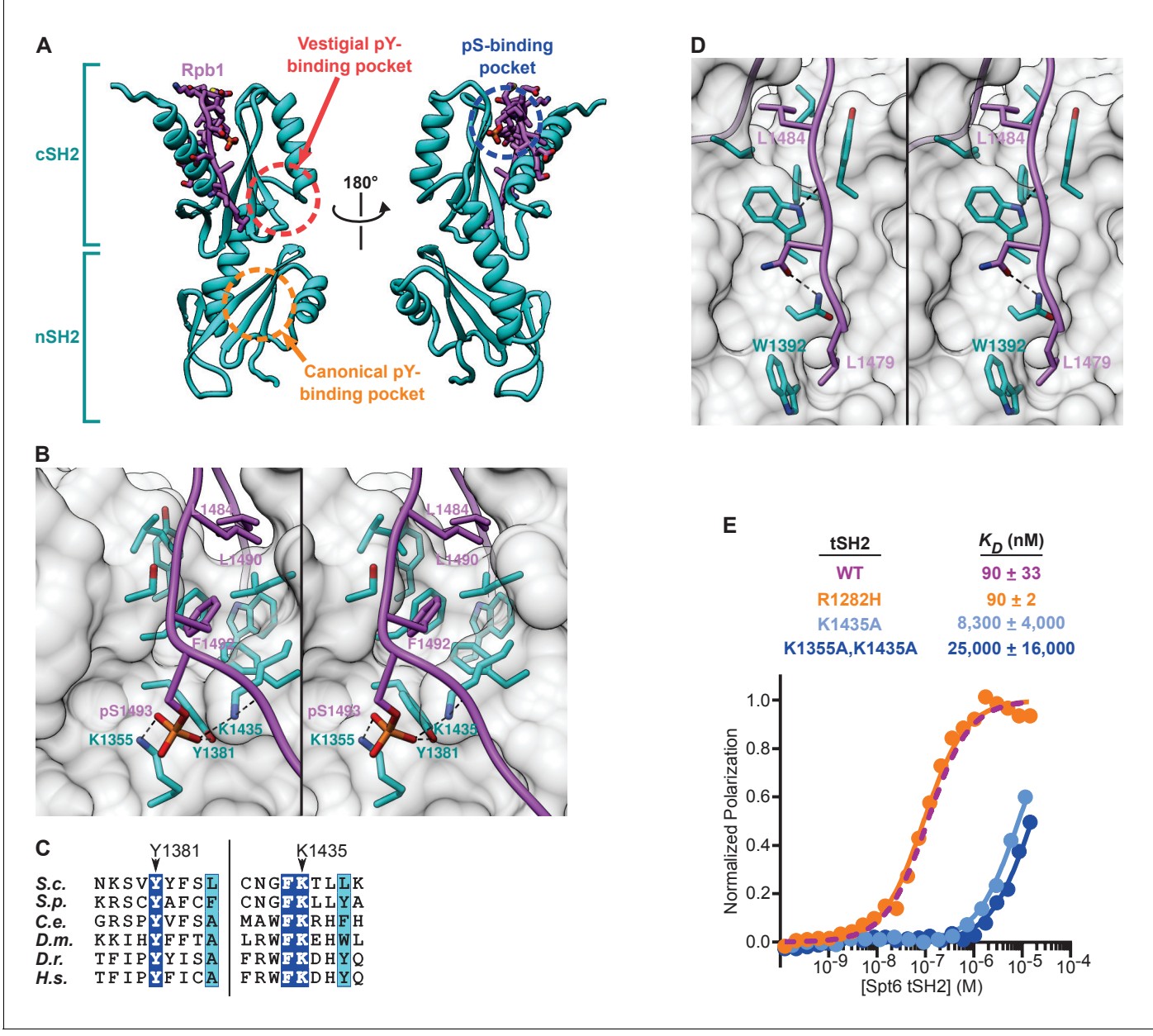

**Figure 3.** Structure of the Spt6 tSH2-Rpb1 linker complex. (**A**) Front and back views. The Spt6 tSH2 nSH2 and cSH2 domains are indicated. (**B**) Stereo view of the pS1493 binding pocket and hydrophobic groove. (**C**) Conservation of Spt6 residues that contact pS1493, as described in *Figure 1E*. Residues contacting the phosphate are indicated (*S. cerevisiae* numbering). Blue background, strictly conserved. Cyan background, highly similar (global similarity score ≥0.7). (**D**) Stereo view of part of the Rpb1 binding groove on the front of Spt6 cSH2. (**E**) Spt6 tSH2 mutant proteins were titrated into the Fl-Rpb1$^{1476-1511\ pS1493}$ peptide and polarization was detected. Average binding affinities calculated from at least 3 independent experiments and estimated standard deviations are shown. The wild type curve from *Figure 2* is reproduced here for reference. See also *Figure 3—source data 1*.

DOI: https://doi.org/10.7554/eLife.28723.006

The following source data is available for figure 3:

**Source data 1.** Binding of fluorescently labeled Rpb1 linker peptides to Spt6 tSH2 mutant proteins, related to *Figure 3E*.

DOI: https://doi.org/10.7554/eLife.28723.007

binding determinants (Fl-Rpb1$^{1468-1500}$). The pS1493 peptide (Fl-Rpb1$^{1468-1500\ pS1493}$) bound with a $K_D$ of 113 nM, which is similar to the 90 nM affinity measured for the shorter Fl-Rpb1$^{1476-1511\ pS1493}$ peptide (*Figures 2A*, *4B and C*). Importantly, phosphorylation of either T1471 or Y1473 in combination with pS1493 (Fl-Rpb1$^{1468-1500\ pT1471,pS1493}$ or Fl-Rpb1$^{1468-1500\ pY1473,pS1493}$) enhanced binding

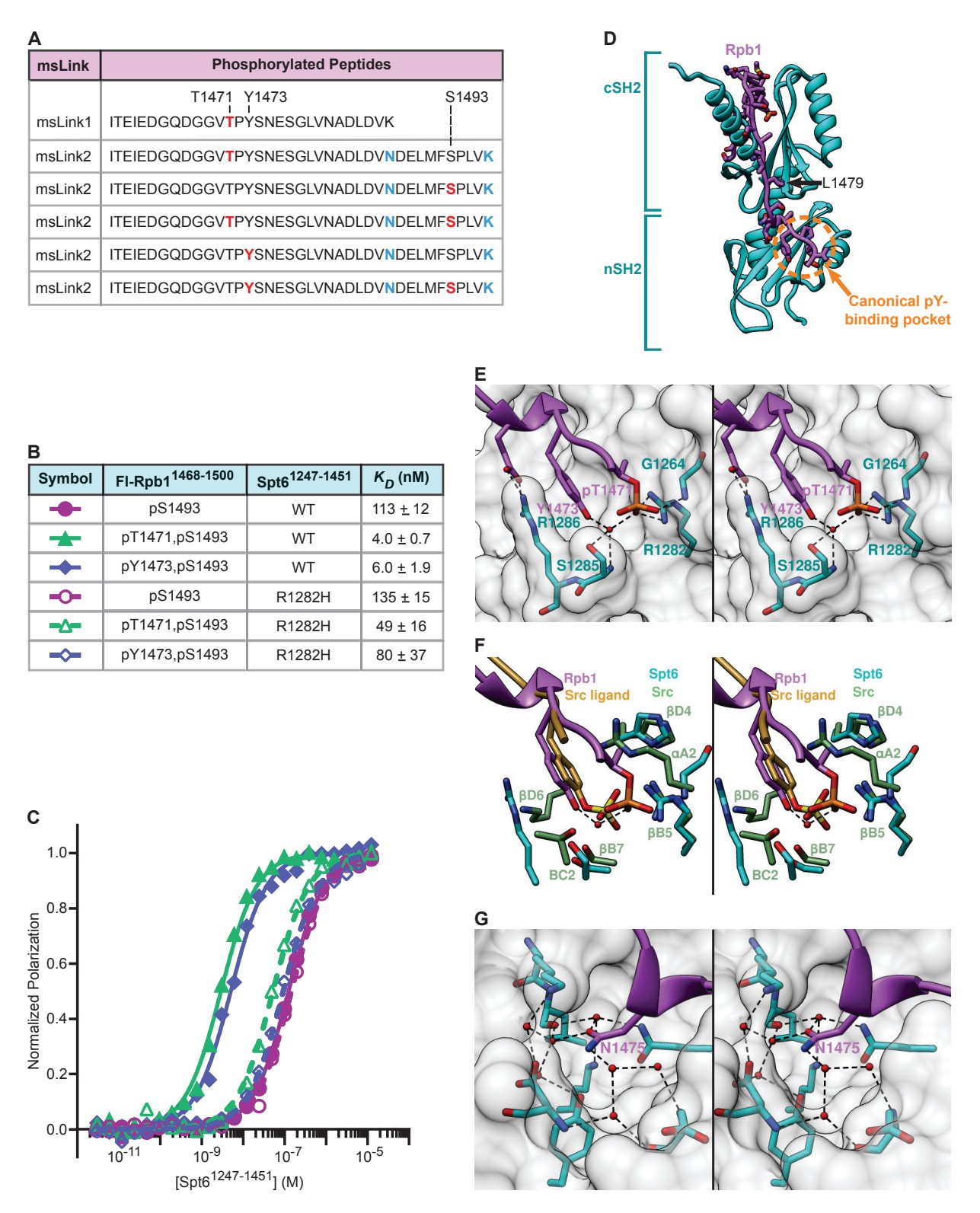

**Figure 4.** Binding of Rpb1 pT1471 and pY1473 to Spt6 tSH2, and mimicry of canonical SH2 interactions. (**A**) Tryptic peptides from msLink1 and msLink2 observed by mass spectrometry, with phosphorylated sites indicated in red, and mutated residues in blue. (**B**) Spt6 tSH2 (wild type or R1282H) was titrated into Fl-Rpb1$^{1468-1500\ pS1493}$, Fl-Rpb1$^{1468-1500\ pT1471,pS1493}$, or Fl-Rpb1$^{1468-1500\ pY1473,pS1493}$ peptides and polarization was detected. Average binding affinities calculated from at least 3 independent experiments and estimated standard deviations are shown. (**C**) Fluorescence polarization-

*Figure 4 continued on next page*

*Figure 4 continued*

derived binding curves from B. (D) Spt6 tSH2-Rpb1$^{1468-1500 \ pT1471,pS1493}$ linker structure. L1479 is the first ordered residue in the pS1493 structure of *Figure 3*, and is indicated with an arrow. (E) Stereo view of the Spt6 tSH2 pT1471-binding pocket. An ordered water molecule is shown as a red sphere. (F) Stereo view of the Spt6 tSH2 pT-binding pocket superimposed with the pY-binding pocket from the Src SH2 domain (PDP: 1SPS). (G) Stereo view of the Spt6 tSH2 specificity pocket. Ordered water molecules are shown as red spheres. See also *Figure 4—figure supplement 1*, *Videos 1–5*, and *Figure 4—source data 1*.

DOI: https://doi.org/10.7554/eLife.28723.009

The following source data and figure supplement are available for figure 4:

**Source data 1.** Binding of fluorescently labeled, doubly phosphorylated Rpb1 linker peptides to Spt6 tSH2 proteins, related to *Figure 4B and C*.

DOI: https://doi.org/10.7554/eLife.28723.011

**Figure supplement 1.** Mass spectra of phosphorylated Rpb1 linker peptides identified by mass spectrometry.

DOI: https://doi.org/10.7554/eLife.28723.010

28-fold (4.0 nM) and 19-fold (6.0 nM), respectively (*Figure 4B and C*). The singly phosphorylated pY1473 peptide (Fl-Rpb1$^{1468-1500 \ pY1473}$) did not bind the tSH2 domain (up to 10 µM tSH2; data not shown), indicating that pY1473 is less important for binding than pS1493. Because pT1471 contributes similar binding energy as pY1473 in the doubly phosphorylated peptides, we do not expect the singly phosphorylated pT1471 peptide to bind either, although we have not tested this directly.

To determine if pT1471 and pY1473 binding involves the canonical pY-binding pocket of the nSH2 domain, we quantified binding to an Spt6$^{R1282H}$ mutant protein (Spt6 R1282 is analogous to the βB5 arginine residue that is critical for pY-binding in canonical SH2 domains [*Kuriyan and Cowburn, 1997*]). This mutation caused a 12-fold decrease in affinity for both of the doubly phosphorylated peptides, confirming the importance of conserved features of SH2 binding pockets for the interaction with pT1471 and pY1473 (*Figure 4B and C*). The contribution of these residues to binding detected by fluorescence polarization but not a far western assay (*Figure 1*) is presumably explained by the limitations of the latter assay and/or variable phosphorylation levels for the relevant residues in vivo.

## Recognition of pT1471 by a canonical SH2 domain

The contribution of pT1471 to binding was unexpected because the nSH2 domain closely resembles canonical SH2 domains, which are highly specific for pY. To visualize the basis for this interaction, we determined a crystal structure of Spt6 tSH2 in complex with Rpb1$^{1468-1500 \ pT1471,pS1493}$ (multiple attempts to determine the structure of a complex with Rpb1$^{1468-1500 \ pY1473,pS1493}$ were unsuccessful). The unit cell and space group were the same as the earlier pS1493 complex crystal, and molecular replacement followed by refinement against 1.8 Å data to R$_{work}$/R$_{free}$ values of 16.2%/19.8% revealed a very similar structure, with additional electron density corresponding to the N-terminal

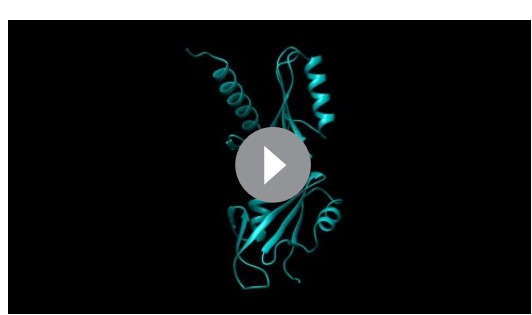

**Video 1.** Overview of the Spt6 tSH2-Rpb1 linker complex. Related to *Figure 4*.

DOI: https://doi.org/10.7554/eLife.28723.012

**Video 2.** Unbiased Fo-Fc omit map around Rpb1. Related to *Figure 4*. The electron density map was phased on a tSH2 model with random 0–0.1 Å shifts applied to every atom and was refined in the absence of the Rpb1 peptide. The base contour level for the Fo-Fc map is 2.5 x RMSD with areas of weaker electron density contoured to 1.8 x RMSD. Density within 2 Å of an atom in the Rpb1 peptide is shown.

DOI: https://doi.org/10.7554/eLife.28723.013

region of the Rpb1 peptide that includes pT1471 (*Figure 4D*; *Videos 1*, *2* and *3*).

Remarkably, the geometry of the bound pTPY$^{1471-1473}$ triad is strikingly similar to the structure of pY bound in canonical SH2 domains (*Figure 4E and F*; *Videos 4* and *5*). The unmodified Rpb1 Y1473 side chain occupies the typical position of the phenolic ring of pY, while the phosphate is instead contributed by pT. The phosphate is displaced 3–3.5 Å from the canonical location, and is hydrogen-bonded to the Y1473 hydroxyl via an ordered water molecule. This allows many of the interactions that stabilize binding in conventional SH2 domain interactions to be conserved in the Spt6-pT complex. For example, the water molecule that bridges the T1471 phosphate and the Y1473 hydroxyl is anchored to Spt6 through hydrogen bonds with the hydroxyl group and main chain nitrogen of S1285, which is analogous to the SH2 BC2 residue (*Kuriyan and Cowburn, 1997*) that in many SH2 domains is a serine or threonine that hydrogen bonds with the tyrosine phosphate. Similarly, Spt6 R1282 (SH2 βB5) hydrogen bonds the phosphate, as it does in canonical SH2-pY interactions. The 3–3.5 Å displacement of the phosphate relative to conventional pY complexes allows formation of an additional hydrogen bond interaction with the main chain carbonyl of Spt6 G1264 (SH2 αA2). In many SH2 domains, the αA2 residue is an arginine or lysine that forms a cation-pi interaction with the pY, so a glycine in this position may open the pocket to allow the displacement of the phosphate seen in our structure. The crystallographic interface therefore explains the binding data, and reveals a striking example of molecular mimicry with canonical SH2-pY interactions.

The specificity of canonical SH2 domains for ligands is also determined by interaction of a ligand side chain +1 to +4 residues after the pY with a specificity pocket adjacent to the pY-binding site. In the Spt6 tSH2 domain, this pocket is occupied by the Rpb1 N1475 side chain, which forms hydrogen bonds with several ordered water molecules that bridge interactions with Spt6 (*Figure 4G*). The main chain carbonyl of N1475 also hydrogen bonds with Spt6, as do the side chains of the two flanking residues S1474 and E1476. Multiple residues in the Rpb1 linker are therefore expected to contribute to the interaction with Spt6 tSH2.

## The Spt6-Rpb1 interaction is important for maintaining repressive chromatin

To determine the importance of Spt6 tSH2 binding to the Rpb1 linker in vivo, we introduced mutations into the native *SPT6* and *RPB1* genes at the normal genomic loci. Mutations affected either the pT/pY interface or the pS interface by altering either the ligand sites (*RPB1* alleles) or the binding pockets (*SPT6* alleles). Individual mutations or combinations were assessed for phenotypes that have been previously reported for tSH2 truncation or deletion alleles, such as Suppressor of Ty (Spt⁻), cryptic intragenic initiation of transcription, and temperature and hydroxyurea sensitivities (*Cheung et al., 2008*; *Lycan et al., 1987*; *McCullough et al., 2015*) (*Figure 5*, *Figure 5—figure supplement 1*). As described below, the pT/pY and pS interfaces both contribute to the maintenance of repressive chromatin.

Neither *spt6* nor *rpb1* mutations that altered only the pT/pY interface caused phenotypes under the conditions tested (*Figure 5B*, *Figure 5—figure supplement 1B*). The *spt6-R1282H* allele (alias = *spt6-R-H*$^{(T/Y)}$ where $^{(T/Y)}$ denotes a mutation in the pT/pY-binding pocket) was described previously (*Close et al., 2011*) and replaces the evolutionarily conserved phosphate-coordinating SH2 βB5 arginine with a histidine. The new Rpb1 mutant alleles *rpb1-T1471A* (alias = *rpb1-T-A*) and *rpb1-Y1473A* (alias = *rpb1-Y-A*) substitute the phosphorylatable T1471 or Y1473 linker residues with alanine. These mutations did not cause a phenotype on their own or in combination with *spt6-R-H*$^{(T/Y)}$, indicating that this interface is not required for optimal growth under the conditions tested. This is consistent with previously published results (*Close et al., 2011*).

In contrast, mutating the pS1493 interface caused the Spt⁻ phenotype, which indicates activation of a weak promoter that is normally repressed by the local chromatin configuration (*Figure 5B*, *Figure 5—figure supplement 1B*, *Figure 5—figure supplement 2A*). The *spt6-K1355A/K1435A* allele (alias = *spt6-KK-AA*$^{(S)}$ where $^{(S)}$ denotes a mutation in the pS-binding pocket) replaces the two lysines that coordinate pS1493 with alanines, and caused a moderate Spt⁻ phenotype. A similar effect was observed with *rpb1-S1493A* (alias = *rbp1-S-A*). The appearance of phenotypes upon mutating the pS interface but not the pT/pY interface is consistent with the greater contribution of pS1493 to binding (*Figure 4B and C*).

Consistent with our in vitro results showing that pS1493 binding was enhanced by pT1471 or pY1473 (*Figure 4B and C*), the effects of mutations that alter the pS1493 interface were enhanced

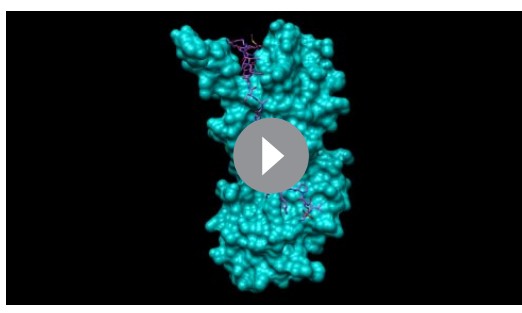

**Video 3.** Interaction network of the Spt6 tSH2-Rpb1 linker interface. Related to **Figure 4**. Water molecules that bridge the Spt6 tSH2-Rpb1 linker interaction are shown as red spheres. Hydrogen bonds identified using UCSF Chimera FindHBond (**Pettersen et al., 2004**) without relaxed constraints are depicted as dashed yellow lines.
DOI: https://doi.org/10.7554/eLife.28723.014

when combined with alterations of the pT1471/pY473 interface (**Figure 5B**, **Figure 5—figure supplement 1B**). The spt6-R-H(T/Y),KK-AA(S), rpb1-T-A,S-A, and rpb1-Y-A,S-A alleles all disrupt both binding interfaces simultaneously, and each caused much stronger phenotypes than the single interface alleles, thereby confirming that the two interactions have overlapping roles in vivo. These strains displayed enhanced Spt⁻ phenotypes (spt6-R-H(T/Y),KK-AA(S) and rpb1-T-A,S-A alleles saturated this reporter) as well as defects in chromatin restoration during transcription elongation. In this assay, activation of a cryptic promoter located within the FLO8 gene drives expression of a HIS3 cassette, allowing cells to grow on media lacking histidine (**Cheung et al., 2008**). With the GAL1 promoter driving FLO8, activation of the HIS3 reporter on galactose media indicates loss of repression under high levels of transcription (**Figure 5—figure supplement 2B**). The spt6-R-H(T/Y),KK-AA(S) and rpb1-T-A,S-A alleles strongly activated this reporter, thereby suggesting defects in transcription-dependent chromatin-mediated repression. The rpb1-Y-A,S-A only weakly activated this reporter, indicating that Rpb1 Y1473 is less important than Rpb1 T1471 for repression of this reporter. Collectively, these data validate the functional importance of the Spt6-Rpb1 interface that we have characterized biochemically and structurally, demonstrate that the pT/pY- and pS-binding pockets collaborate in vivo, and suggest that the Spt6-Rpb1 interaction is important for the reassembly of repressive chromatin during transcription. For a more thorough genetic analysis of mutations that disrupt the Spt6-Rpb1 interaction and an accompanying discussion, please see **Figure 5—figure supplement 1**.

## The Rpb1 linker recruits Spt6 to highly transcribed transcription units

Our data support a model in which specific phosphorylation of the Rpb1 linker recruits Spt6 to sites of transcription where it is needed to reassemble nucleosomes during transcription. To test this model, we measured genome-wide occupancy of Spt6 by ChIP-seq in a series of mutants

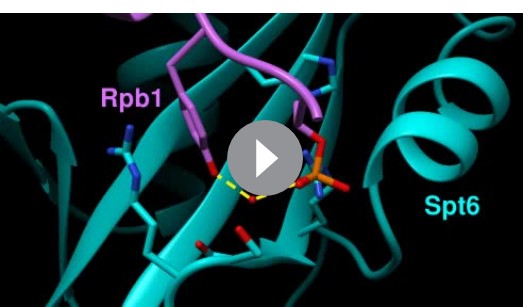

**Video 4.** Comparison of the Spt6 tSH2-Rpb1 pT1471/Y1473 interaction with other SH2-pY interactions. Related to **Figure 4**. Structures of SH2 domains and their pY ligands (Src, PDB: 1SPS; BRDG1, PDB: 3MAZ; CBL-N, PDB: 2CBL; Gads, PDB: 5GJH; Syp, PDB: 1AYA) were aligned on the Spt6 tSH2-Rpb1 linker structure. Alignments were performed on the Cα atoms of conserved residues in the pY-binding pocket (αA2, βB5, βB7, and βD4; the CBL-N alignment also included the Cα atoms of Spt6 R1286 and CBL R299).
DOI: https://doi.org/10.7554/eLife.28723.015

**Video 5.** Structural model of the Spt6 tSH2 interaction with Rpb1 pY1473. Related to **Figure 4**. A phosphate with appropriate geometry was modeled onto the hydroxyl group of Rpb1 Y1473 without any other structural adjustment. Hydrogen bonds are displayed as dashed lines (precise geometric criteria (**Pettersen et al., 2004**), yellow; within 0.4 Å and 20° of precise geometric criteria, purple).
DOI: https://doi.org/10.7554/eLife.28723.016

affecting one or both binding pockets or ligand sites. Consistent with the model and with previous reports (*Burugula et al., 2014*; *Mayer et al., 2010*), we found that the Spt6 occupancy profile matches that of RNAPII, with Spt6 found across the average transcription unit in amounts roughly proportional to the level of transcription (*Figure 6*, *Figure 6—figure supplement 1*). To our surprise, Spt6 or Rpb1 mutations that impair either single phosphate-binding pocket (*spt6-KK-AA$^{(S)}$*, *spt6-R-H$^{(T/Y)}$*, and *rpb1-FSP$^+$-KRR$^+$*; see *Figure 5—figure supplement 1* for a description of the *rpb1-FSP$^+$-KRR$^+$* allele) caused small *increases* in Spt6 occupancy over gene bodies. In contrast, the more severe Spt6 or Rpb1 mutations that impair both phosphate-binding pockets (*spt6-R-H$^{(T/Y)}$,KK-AA$^{(S)}$*, *rpb1-T-A,FSP$^+$-KRR$^+$*, and *rpb1-Y-A,FSP$^+$-KRR$^+$*) each caused significant *decreases* in Spt6 occupancy, supporting the model that the binding detected in vitro enhances recruitment of Spt6 to sites of RNAPII activity. Notably, the strongest effects of mutations in the Spt6-Rpb1 linker interface occurred in genes with the highest transcription levels (*Figure 6*, *Figure 6—figure supplement 1*). Overall, the ChIP-seq data support the model that the interactions measured in vitro are important for recruiting Spt6 to sites of transcription, especially sites that are heavily transcribed, but they also suggest that this phosphorylation-dependent switch has a complex output (discussed below).

## Discussion

Prior to this study, it was generally assumed that Spt6 binds to the phosphorylated heptad repeats of the RNAPII CTD (*Yoh et al., 2007*; *Yoh et al., 2008*). That model was made attractive by the established role of CTD phosphorylations in binding RNAPII cofactors (*Eick and Geyer, 2013*), the presence of pY residues in the heptad repeats (*Mayer et al., 2012*), and the numerous characterized interactions of SH2 domains with pY residues (*Machida and Mayer, 2005*). In contrast to this assumption, we found that the Spt6 tSH2 domain binds to the flexible Rpb1 linker that tethers the CTD heptad repeats to the core enzyme rather than to the CTD repeats themselves, thereby uncovering a functional role of the Rpb1 linker. Phosphorylation of Rpb1 residues T1471, Y1473, and S1493 within this linker contributed to tight binding, and crystal structures revealed unexpected mechanisms for recognition of pT1471 and pS1493, while biochemical data imply a canonical SH2 interaction for pY1473. Rpb1 pT1471 and Y1473 collaborated to bind the nSH2 pocket in a manner mimicking canonical pY binding, and Rpb1 pS1493 bound a novel pS-binding pocket that is located 70 Å distant. We further demonstrated that Rpb1 T1471, Y1473, and S1493 are phosphorylated in vivo, validated the interface biochemically and genetically, and found that the interface is important for the recruitment of Spt6 to highly transcribed genes and for the maintenance of repressive chromatin.

Mutations that disrupt the Spt6 tSH2-Rpb1 linker interaction caused defects in transcription-dependent chromatin-mediated repression of cryptic promoters, as indicated by appearance of the Spt$^-$ phenotype and the activation of a cryptic promoter reporter (*Figure 5*). These observations indicate that maintaining the appropriate association of Spt6 with RNAPII is important for maintaining a repressive chromatin structure, which is consistent with a role for Spt6 in the reassembly of nucleosomes in the wake of elongating RNAPII, an activity that is likely mediated by interactions of the Spt6 N-terminal domain with nucleosomes/histones (*Duina, 2011*; *McDonald et al., 2010*).

Deletion of the tSH2 domain has been previously shown to greatly reduce Spt6 occupancy at a subset of protein coding genes (*Burugula et al., 2014*; *Mayer et al., 2010*). We find that mutations that impair both of the phosphate-binding pockets in the Spt6 tSH2-Rpb1 linker interface also reduce Spt6 occupancy at the average gene, with large effects at frequently transcribed genes. This suggests that Rpb1 T1471/Y1473 and S1493 phosphorylation-dependent recruitment of Spt6 is utilized more frequently at highly transcribed genes. An attractive model is that engaging a dedicated molecule of Spt6 during elongation is helpful, but is not essential for RNAPII transcription, possibly because the high abundance of Spt6 (about one molecule per three nucleosomes or ~10 μM [*McCullough et al., 2015*]) allows the essential histone chaperone functions of Spt6 to be performed with diminished but sufficient efficiency through spontaneous encounters. At highly transcribed genes, where there is increased disruption of chromatin due to the more frequent passage of RNAPII, a dedicated Spt6 has to be recruited to the phosphorylated Rpb1 linker to maintain an appropriate chromatin state. This is consistent with a study that found greater loss of H3 at highly transcribed genes in the absence of Spt6 (*Perales et al., 2013*).

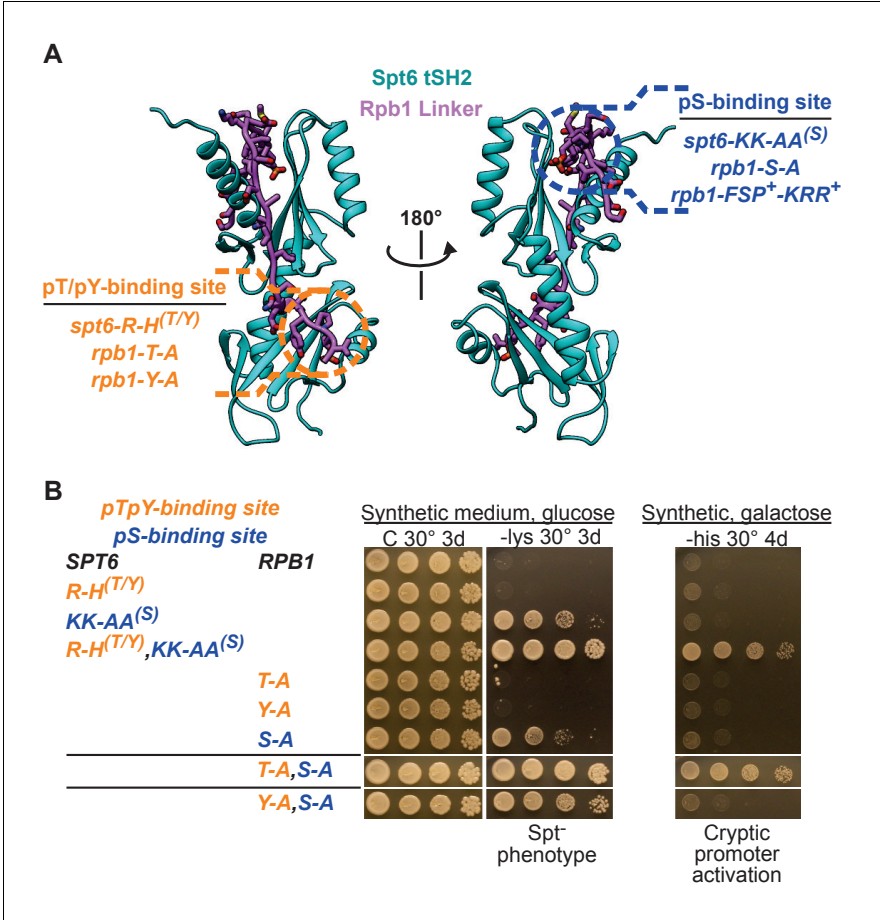

**Figure 5.** The Spt6-Rpb1 interaction is important for maintaining repressive chromatin. (**A**) Structural map of Spt6 and Rpb1 mutations that were integrated at their native loci. For a description of the *rpb1-FSP⁺-KRR⁺* allele and its phenotypes, refer to *Figure 5—figure supplement 1*. (**B**) Growth conditions/phenotype (above/below) for the mutations shown in A. Growth on media lacking lysine indicates the Spt⁻ phenotype and growth on galactose media lacking histidine indicates activation of the cryptic promoter within *FLO8* with high levels of transcription. See also *Figure 5—figure supplement 1* and *Figure 5—figure supplement 2*.
DOI: https://doi.org/10.7554/eLife.28723.017

The following figure supplements are available for figure 5:

**Figure supplement 1.** Complete genetic analysis of mutations that disrupt the Spt6-Rpb1 interaction.
DOI: https://doi.org/10.7554/eLife.28723.018

**Figure supplement 2.** Schematic of Spt⁻ and cryptic promoter activation reporters used to assay yeast phenotypes.
DOI: https://doi.org/10.7554/eLife.28723.019

In contrast to the severe Spt6 and Rpb1 mutants, less severe mutations that impair a single phosphate-binding site increase Spt6 occupancy across gene bodies. One possible explanation for this observation is that these less severe mutations allow relatively normal recruitment of Spt6 to transcription sites, but cause a defect in RNAPII progression that results in increased RNAPII (and therefore Spt6) retention/occupancy. Our data also include the intriguing observation that Spt6 occupancy is skewed toward the 3' end of average genes, and that the accumulation of Spt6 directly over the transcription termination site is diminished or lost in our binding interface mutants (*Figure 6C*), which may reflect a role for this interaction in termination of transcription. Definitive explanations for these various correlations will require further study.

As noted above, Spt6 is highly abundant in the yeast nucleus (~10 μM [*McCullough et al., 2015*]), which raises interesting questions about the affinities we observed for biochemical binding of phosphorylated peptides. In particular, the 100 nM affinity of the singly phosphorylated pS1943 peptide

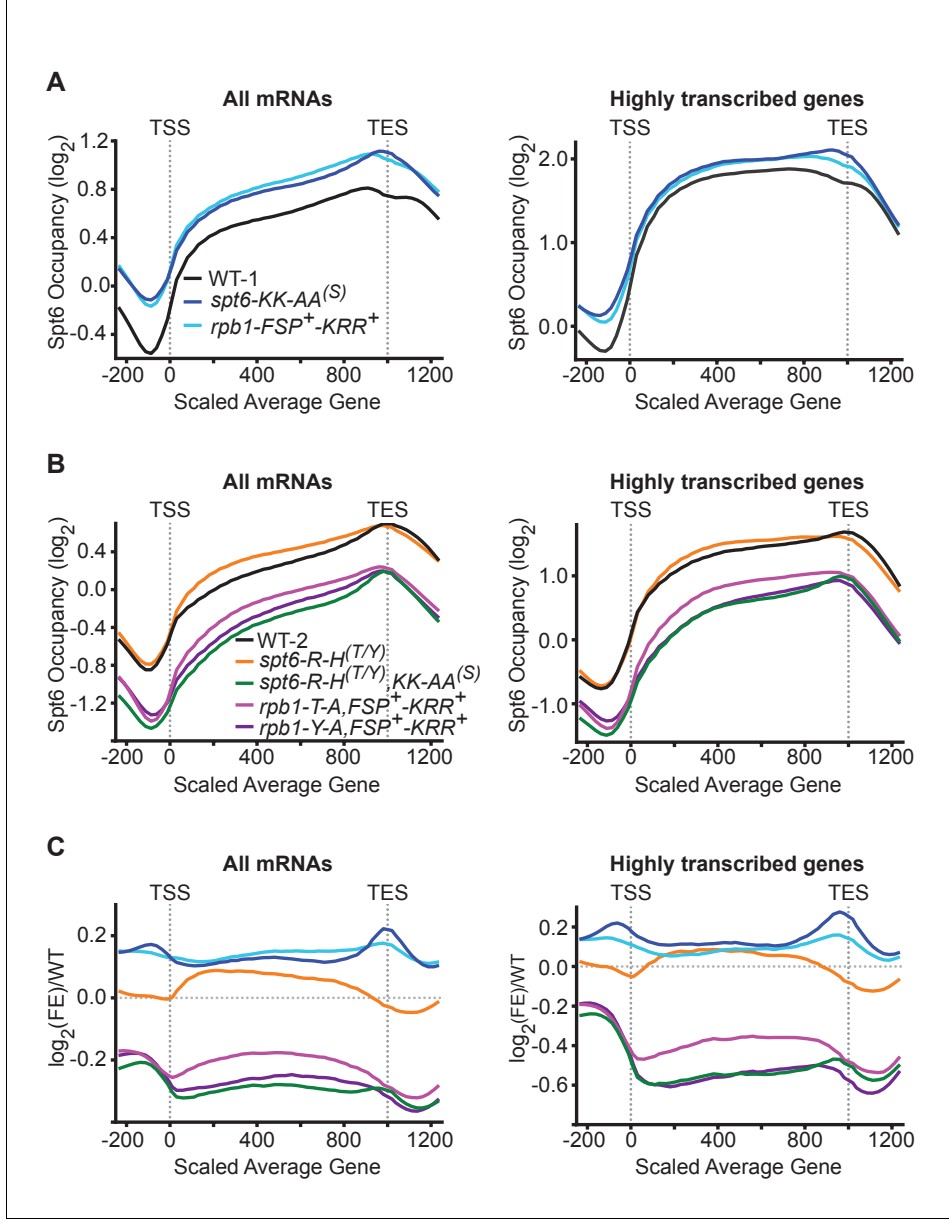

**Figure 6.** The Spt6-Rpb1 interaction is important for maintaining Spt6 occupancy at transcribed genes. (**A**) The log$_2$ enrichment ratios of Spt6 across all protein coding genes (left) and the top 20% of most frequently transcribed genes (right) in wild type yeast or yeast harboring *spt6-KK-AA*$^{(S)}$ or *rpb1-FSP*$^+$*-KRR*$^+$ mutant alleles (see *Figure 5* and *Figure 5—figure supplement 1* for allele designations). −200 to 0 and 1000–1200 are in base pairs, 0–1000 is normalized to the average transcript size. (**B**) As in A except mutant alleles include *spt6-R-H*$^{(T/Y)}$, *spt6-R-H*$^{(T/Y)}$*,KK-AA*$^{(S)}$, *rpb1-T-A,FSP*$^+$*-KRR*$^+$, or *rpb1-Y-A,FSP*$^+$*-KRR*$^+$. (**C**) The log$_2$ enrichment ratios of Spt6 in panels A and B normalized to the respective wild type occupancy. See also *Figure 6—figure supplement 1* and *Figure 6—figure supplement 2*.

DOI: https://doi.org/10.7554/eLife.28723.020

The following figure supplements are available for figure 6:

**Figure supplement 1.** Mutations that impair the Spt6-Rpb1 linker interface reduce Spt6 occupancy at highly transcribed genes.

DOI: https://doi.org/10.7554/eLife.28723.021

**Figure supplement 2.** Antibody raised against *S. cerevisiae* Spt6 cross-reacts with *C. glabrata* Spt6.

DOI: https://doi.org/10.7554/eLife.28723.022

might be sufficient to ensure saturation of Spt6-Rpb1 binding in the absence of a reinforcing interaction from phosphorylation at T1471 or Y1473. One possibility is that the Spt6 tSH2 domain has additional ligands in the cell, and that T1471 or Y1473 phosphorylation is required to drive Rpb1 association in the face of competing interactions.

In addition to the maintenance of repressive chromatin, Spt6 is reported to participate in mRNA processing and export, although the mechanistic role of Spt6 in these processes is unknown (*Andrulis et al., 2002*; *Estruch et al., 2009*; *Yoh et al., 2007*). One possibility is that Spt6 forms a bridge between the transcription machinery and mRNA processing and export factors. In support of this, the N-terminus of Spt6 directly binds the factor Spn1 (*Diebold et al., 2010a*; *McDonald et al., 2010*; *Yoh et al., 2007*), which has been implicated in these processes. One attractive model is that the interaction between the Spt6 tSH2 domain and the Rpb1 linker might recruit Spt6 to transcription complexes where it serves as a binding platform for mRNA processing and export factors such as Spn1. Thus, it is possible that the interaction that we have characterized with the Rpb1 linker might contribute to multiple known functions of Spt6, and testing these possibilities and associated mechanisms will be a priority for future studies.

Human SPT6 was found to associate with an Rpb1 construct that had been incubated with the kinase P-TEFb (*Yoh et al., 2007*; *Yoh et al., 2008*). The role of P-TEFb in yeast is typically performed by the cyclin-dependent kinases Ctk1 and Bur1 (*Bowman and Kelly, 2014*). Consistent with a possible role for Ctk1 and/or Bur1, T1471 and P1472 comprise a minimal cyclin-dependent kinase site and the residues flanking Rpb1 S1493 resemble those of known Ctk1 and Bur1 substrates. Moreover, Ctk1 and Bur1 have been physically and functionally linked to Spt6, and contribute to Spt6 localization at the 5' end of genes (*Burugula et al., 2014*; *Dronamraju and Strahl, 2014*). In contrast, candidate kinases for phosphorylating Y1473 are less obvious because there are no dedicated tyrosine kinases recognized in *S. cerevisiae*. Identification of the authentic physiological kinase(s) and the timing and regulation of their T1471, Y1473, and S1493 phosphorylation activities are important goals of future studies.

The tSH2-pT1471 structure provides a remarkable example of molecular mimicry in which the components of a cognate pY ligand are provided in two parts: a tyrosine side chain (Y1473) and a phosphate (pT1471). Because the Spt6 tSH2 represents the only SH2 domain that has been recognized in yeasts and a dedicated yeast tyrosine kinase has not been described, it is attractive to speculate that the Spt6 tSH2 domain reflects a primordial interaction module that bound pT-X-Y peptides (*Liu and Nash, 2012*). The subsequent emergence of a tyrosine kinase may have led to an altered signaling pathway in which the molecular recognition responded to phosphorylation of the tyrosine. Thus, our structural and biochemical observations suggest a plausible evolutionary origin of the highly abundant and diverged SH2-pY interaction family of metazoa. A related concept that merits future study is that the ability of alternative pT or pY to bind an SH2 site might allow distinct kinases to converge on the same molecular interaction, thereby further enriching the potential of many signaling pathways.

## Materials and methods

### Plasmids

Plasmids used in this study are described in *Supplementary file 1*. For plasmids expressing mutant Rpb1, PCR fragments containing the desired mutation were generated using overlap extension PCR as described previously (Saiki et al, Nucleic Acid Research, 1988) and ligated into the Sph1 and Xho1 sites of pCK859 (*Suh et al., 2010*). Plasmids expressing GST-tagged Spt6 were produced by inserting the Spt6$^{1223-1451}$ coding sequence into the pDEST15 vector (Thermo Fisher Scientific, Waltham, MA). Mutations in the tSH2 domain were introduced using site-directed mutagenesis.

### Saccharomyces cerevisiae

Yeast strains used in this study are described in *Supplementary file 2*. Mutations in *SPT6* and *RPB1* were integrated into the endogenous loci in strains congenic with A364a by PCR-mediated transformation (*McCullough et al., 2015*) and combined using standard yeast genetic methods.

## Protein expression and purification

Wild type endogenous *S. cerevisiae* RNAPII was purified from yeast strain MAS015 containing a PreScission Protease cleavable Protein A tag at the C-terminus of the Rpb3 subunit. RNAPII ΔCTD, Δlinker, and Rpb1 point mutants were purified from 9138-4-2, a derivative of yeast strain CKY283 (*Kaplan et al., 2008*; *Larson et al., 2012*) containing a TAP tag at the C-terminus of the Rpb3 subunit and transformed with mutated versions of *RPB1* on a plasmid covering a deletion of this gene. RNAPII was purified from 6 liters of YPD (yeast extract peptone dextrose) inoculated with 100 mL of a saturated overnight culture and incubated at 30°C for 2 days. Cells were harvested by centrifugation, washed once with cold water, and pelleted by centrifugation. Cells were frozen by passing through a syringe into liquid nitrogen and lysed under liquid nitrogen using a SPEX SamplePrep 6870 Freezer/Mill (SPEX SamplePrep, Metuchen, NJ). Pulverized yeast were thawed in 1 pellet equivalent of lysis buffer (50 mM Tris-Cl pH 7.5, 500 mM KCl, 10% glycerol, 0.1% Tween-20, 10 µM $ZnCl_2$, 1 mM DTT, 1.4 µg/mL pepstatin, 1 µg/mL leupeptin, 1 µg/mL aprotinin, 1.9 mM PMSF). Lysates were clarified by centrifugation at 37,000 RCF for 30 min. The supernatant was incubated with 2 mL of IgG sepharose resin (MP Biomedicals Cappel[TM] antigen affinity gel, Santa Ana, CA) for 1 hr, followed by 6 washes with 4 mL of lysis buffer. Proteins were eluted by incubating with 4 mL of lysis buffer containing 100 µg PreScission Protease (MAS015; PreScission Protease-Protein A tag) or TEV Protease (9138-4-2 derived strains; TAP tag) overnight at 4°C. The subsequent flow-through fraction was collected and the resin was washed with 4 mL of lysis buffer. The flow through and wash were pooled and diluted 4-fold with buffer QA (20 mM Tris-acetate pH 7.5, 0.5 mM EDTA, 10 µM $ZnCl_2$, 10% glycerol, 1 mM DTT) followed by anion exchange chromatography (5 mL Hi-Trap Q column; GE Healthcare Life Sciences, Chicago, IL) using a gradient from 150 mM to 1,500 mM KOAc. The RNAPII peak fractions were collected, concentrated to 10 mL, and diluted 2-fold with buffer QA to reduce the salt concentration. RNAPII was exchanged into storage buffer (25 mM HEPES pH 7.5, 150 mM KOAc, 10 µM $ZnCl_2$, 10% glycerol, 5 mM DTT) by serial concentration and dilution. RNAPII was concentrated to ~5 µM and flash frozen in liquid nitrogen prior to storage at −80°C.

Spt6[1247-1451] proteins were expressed and purified as described previously (*Close et al., 2011*). GST-Spt6[1223-1451] expression and purification was the same with the following modifications. Lysates were incubated for 3 hr with Pierce glutathione agarose resin (Thermo Fisher Scientific) in GST lysis buffer (50 mM Tris-Cl pH 7.5, 500 mM NaCl, 10% glycerol, 0.5 mM EDTA pH 8.0, 1 mM DTT). Beads were washed with 10-column volumes of GST lysis buffer followed by 5-column volumes of GST wash buffer (50 mM Tris-Cl pH 7.5, 100 mM NaCl, 5% glycerol, 0.5 mM EDTA pH 8.0, 1 mM DTT). GST-Spt6[1223-1451] was eluted using 6 column volumes of GST wash buffer supplemented with 10 mM glutathione and further purified as described for Spt6[1247-1451].

For purification of msLink1 or msLink2, 6 L of a yeast strain derived from CKY283 expressing Rpb1-msLink1/2 was grown at 30°C to an OD[600] of 3.5 and harvested as described above. 25 g of cells were suspended in 12.5 mL msLink lysis buffer (50 mM Tris-Cl pH 7.5, 750 mM NaCl, 10% glycerol, 0.1% Tween-20, 10 µM $ZnCl_2$, 1 mM DTT, 0.7 µg/mL pepstatin, 0.5 µg/mL leupeptin, 0.5 µg/mL aprotinin, 960 µM PMSF, 2 mM NaF, 2 mM NaVO$_4$, 2 mM β-Glycerophosphate (BGP)) and frozen as drops in liquid nitrogen. Frozen cells were lysed under liquid nitrogen using a SPEX SamplePrep 6870 Freezer/Mill, rehydrated in 38 mL msLink lysis buffer, and clarified by centrifugation at 37,000 RCF for 30 min. The supernatant was incubated with 1 mL of IgG sepharose resin (MP Biomedicals Cappel antigen affinity gel) for 1 hr, followed by 5 washes with 4 mL of msLink lysis buffer and 3 washes with msLink cleavage buffer (50 mM Tris-Cl pH 7.5, 300 mM NaCl, 10% glycerol, 0.1% Tween-20, 15 mM imidazole, 0.7 µg/mL pepstatin, 0.5 µg/mL leupeptin, 0.5 µg/mL aprotinin, 960 µM PMSF, 2 mM NaF, 2 mM NaVO$_4$, 2 mM BGP). msLink1/2 was eluted by incubating with 1 mL of msLink cleavage buffer containing 50 µg PreScission Protease for 4 hr at 4°C. Following cleavage, the flow through was collected and the resin washed with 2 mL of msLink cleavage buffer without DTT. The flow through and wash were pooled and incubated with 50 µL Ni-NTA agarose (Qiagen, Germantown, MD) for 30 min at 4°C. The beads were washed 3 times with 200 µL msLink cleavage buffer without DTT followed by an additional 3 washes with Ni buffer (25 mM Tris-Cl pH 7.5, 50 mM NaCl, 30 mM imidazole). msLink1/2 was eluted with 1 mL Ni buffer containing 300 mM imidazole and concentrated to 60 µL using a 10 kDa molecular weight cutoff Vivaspin concentrator (Sartorius Stedim Biotech, Aubagne, France).

## LC/MS/MS analysis of peptides from msLink1 and msLink2

Peptides digested with trypsin were analyzed using a nano-LC/MS/MS system equipped with a nanoLC Ultra-2D HPLC pump (Eksigent, Dublin, CA) and a maXis II ETD mass spectrometer (Bruker Daltonics, Bremen, Germany). The maXis II ETD mass spectrometer was equipped with a captive spray ion source. Approximately 5 µL of peptide samples were injected onto an Atlantis dC18 nano-bore LC column (100 µm ID x 100 mm length, 3 mm particles; made in house; Waters Corp, Milford, MA). Peptides were separated and eluted over a linear gradient of 5–96% acetonitrile in 0.1% formic acid with a constant total flow rate of 400 nL/minute over 78 min.

## Protein ID and database searches

Identified peptides were assigned to the msLink1 and msLink2 sequences by searching against a custom database using the MASCOT search engine (in-house licensed; ver. 2.5; Matrix Science, Inc., Boston, MA; RRID:SCR_014322). Molecular ions with +2, +3, or +4 charge states determined from a Q-ToF primary mass spectrum were usually considered. Searches were performed for trypsin-specific cut sites allowing for two missed cleavages and mass error tolerances of 11 ppm for both the Q-ToF primary mass spectrum and the MS/MS ions. Mass data peaklists for the Mascot searches were generated using Compass DataAnalysis V 4.4 software (Bruker Daltonics). Peptide modifications included in the searches were oxidation on methionine and/or phosphorylation on Ser, Thr, or Tyr residues. Peptide assignments were accepted when the MASCOT ion score value was greater than 20, mass errors were less than 11 ppm, and the expect values were less than 1. Phosphopeptide assignments were manually validated.

## Peptide synthesis

Peptides were synthesized on a *Prelude X* instrument (Protein Technologies, Inc., Tucson, AZ) using standard Fmoc solid-phase peptide synthesis at 25 µmole scale. Synthesis-grade Fmoc amino acid (AA) reagents were purchased from Protein Technologies, Inc. or AAPPTec (Louisville, KY). Standard coupling conditions comprised addition of 0.65 mL Fmoc-AA (200 mM in DMF), 0.65 mL HATU (195 mM in DMF), and 0.5 mL NMM (0.6 M in DMF). Coupling reactions were mixed for 25 min at 50℃. Fmoc deprotection employed 20% piperidine in DMF. To generate C-terminal carboxamides, peptides were prepared on TentaGel R RAM resin (Rapp-Polymere, Tübingen, Germany). N-terminal fluorescein capping was performed with 5 (6)-carboxyfluorescein (Sigma-Aldrich, St. Louis, MO) while acetyl capping utilized acetic anhydride with 0.6 M NMM. Peptide resin was thoroughly washed with DCM and dried under vacuum after synthesis was complete. Cleavage of the peptide from resin was performed with 4 mL [92.5/2.5/2.5/2.5%] of TFA/water/TIS/EDT for 3 hr. The TFA solution was then precipitated in ice-cold ether and centrifuged at 5100 RPM. Supernatant was decanted while pellets were washed twice more and then dried overnight under vacuum.

## Generation of phosphorylated residues

To generate phosphorylated residues, peptides were first synthesized with trityl protected Ser, Thr, and/or unprotected Tyr. Trityl protection was removed by exposure to 0.1% TFA in DCM for 6 min. The resin was dried thoroughly in a fritted vessel overnight under vacuum, then sealed and flushed with argon. Five mL of anhydrous DMF were added to reswell the resin while maintaining anhydrous conditions. Five mmol of ethylthiotetrazole in anhydrous DMF were transferred to a dry flask containing 2 mmol dibenzyl N,N-diisopropyl-phosphoramidite. The tetrazole-amidite mixture was immediately transferred to the resin vessel and agitated for 1 hr. After draining and washing the reaction mixture from the resin, 0.5 mL of 3 M t-butyl hydroperoxide in isooctane was added to the resin for 30 min under agitation. Peptide resin was then thoroughly washed with DCM and dried under vacuum. Cleavage and drying of the peptide from resin was performed as above.

## Peptide purification and MS analysis

Crude peptides were dissolved in 6 M Guanidine HCl, 40 mM triethylammonium acetate (TEAA), 20 mM DTT and adjusted to pH 8 with $NH_4HCO_3$. Purification was performed on a Beckman Coulter System Gold HPLC with a 250 × 20 mm Higgins Proto 200 Å, C18, 10 µm column. Mobile phase TEAA buffer (Buffer A: 5% acetonitrile, 40 mM TEAA, pH 7.0; Buffer B: 80% acetonitrile, 40 mM TEAA, pH 7.0) was used at a flow rate of 7.5 mL/minute over an 80 min 10% to 30% gradient with

Buffer B. Fractions were collected over 1 min intervals. Based on UV absorbance at 214 nm, peptide-containing fractions were analyzed further by mass spectrometry. LC/MS analysis was performed on an Agilent 1200 series HPLC with a 4.6 × 50 mm Poroshell 120 Å, EC-C18, 2.7 µm column upstream of an Agilent 6110 Single Quadrupole system (Agilent Technologies, Santa Clara, CA). Mobile phase (Buffer A: 0.1% formic acid in water; Buffer B: 0.1% FA in acetonitrile) was run at 1.0 mL/minute over an 8 min 5% to 90% gradient with Buffer B. Mass spectrometric data were collected by an Agilent 6110 Single Quadrupole with a detection window of 400 to 2000 m/z in positive ion mode. Peptide fractions identified by mass spectrometry and determined to be >95% pure were lyophilized until dry and stored at −80°C.

## Far western blots

Far western blots were performed as described previously (*Wu et al., 2007*). Briefly, 2.3 pmol of purified RNAPII was separated by SDS-PAGE on Invitrogen NuPAGE Novex 4–12% Bis-Tris Protein Gels (Thermo Fisher Scientific) electrophoresed in MOPS running buffer, transferred to nitrocellulose membranes, and denatured in guanidine HCl. Proteins were renatured by stepwise incubation with protein binding buffer (20 mM Tris-Cl pH 7.5, 100 mM NaCl, 0.5 mM EDTA, 10% glycerol, 0.1% Tween-20, 2% skim milk powder, 1 mM DTT) containing decreasing concentrations of guanidine HCl and incubated with blocking buffer (3% milk in TBS (25 mM Tris-Cl, 137 mM NaCl, 2.7 mM KCl) for 1 hr at room temperature. Membranes were incubated with protein binding buffer containing 35 nM GST or GST-Spt6$^{1223-1451}$ overnight at 4°C. Membranes were washed three times with TBS-T (TBS with 0.05% Tween-20) and incubated with anti-GST antibody produced in goat (GE Healthcare 27-4577-01; RRID:AB_771432) diluted 1:1000 in blocking buffer. After washing three times with TBS-T, membranes were incubated with a 1:10,000 dilution of infrared-labeled antisera that recognizes goat IgG (LI-COR, Lincoln, NE; RRID:AB_621846). Bound proteins were detected using an Odyssey scanner (LI-COR). Quantification of band intensity was performed using ImageJ (RRID:SCR_003070).

## Western blots

Yeast whole cell extracts from strain 9138-4-2 transformed with plasmids carrying either wild type Rpb1 or Rpb1 S1493A were prepared using the trichloroacetic acid (TCA) method as described previously (*McCullough et al., 2015*). Protein concentrations were determined using the Pierce BCA Protein Assay Kit (Thermo Fisher Scientific), and 1.6 µg total protein was separated by SDS-PAGE. Rpb1 was detected with primary antibodies y-80 (Santa Cruz Biotechnology, Dallas, TX; RRID:AB_655813) or 8WG16 (Covance, Princeton, NJ; RRID:AB_10013665) that recognize the N-terminal region or CTD respectively and infrared-labeled secondary antibodies (LI-COR; RRID:AB_621848 or RRID:AB_621840).

Polyclonal antiserum that specifically recognizes Rpb1 pS1493 was produced by Covance (RRID: AB_2687438). Briefly, rabbits UT743 and UT744 were injected with the Rpb1 linker-derived peptide CDVKDELMF(pS)PLVDSGSN conjugated to KLH. The exsanguination bleed was subjected to positive and negative affinity purification steps over columns consisting of the phosphorylated peptide or unphosphorylated peptide, respectively. The specificity for the phosphorylated peptide versus the unphosphorylated peptide was validated by dot blot.

For dot blots, 1 µl of a 2-fold serial dilution series starting at 75 µM Rpb1$^{1476-1511}$ were spotted on a nitrocellulose membrane. Peptides were either phosphorylated on S1493, unphosphorylated, or had S1493 substituted to an alanine. The membrane was dried before blocking with 5% BSA and detecting with α-Rpb1 pS1493 primary antibody and infrared-labeled secondary antibody (LI-COR; RRID:AB_10796098).

## Fluorescence polarization

For direct binding assays, purified Spt6$^{1247-1451}$ was exchanged into polarization buffer (15 mM Tris-Cl pH 7.5, 100 mM NaCl, 5% glycerol, 0.5 mM EDTA pH 8.0, 2 mM 2-mercaptoethanol) by size exclusion chromatography or overnight dialysis and titrated in 1.7- or 2-fold serial dilutions starting at 10–15 µM against a constant concentration (0.3 nM) of fluorescently labeled peptide. Reactions were incubated for 5 min at 25°C and fluorescence polarization was measured using a BioTek Synergy Neo Microplate Reader (Winooski, VT) set to 485 nm/535 nm excitation/emission wavelengths. $K_D$ values were determined by fitting the data in GraphPad Prism 7 (GraphPad Software, Inc., La

Jolla, CA; RRID:SCR_002798) using non-linear least squares regression against P = (P$_T$ × (([pro] + [pep]+$K_D$) − (([pro] + [pep]+$K_D$)$^2$ − 4 × [pro] × [pep])$^{1/2}$)) / (2 × [pep]) (*LiCata and Wowor, 2008*), where P is the measured polarization, P$_T$ is the total change in polarization, [pep] is the peptide concentration, and [pro] is the protein concentration. When saturation was not reached, the $K_D$ was estimated by fitting the data under the assumption that saturation would occur at the upper baseline of the binding curves that did reach saturation. Average $K_D$s (mean) and standard deviations were calculated from at least three independent experiments, with independent experiments defined as experiments that were performed with separate protein titrations and buffer preparations.

## X-ray crystallography

For the Spt6$^{1247-1451}$-Rpb1$^{1476-1500\ pS1493}$ complex, 86 µM Spt6$^{1247-1451}$ was mixed with a 2-fold molar excess of Rpb1$^{1476-1500\ pS1493}$ and concentrated to 500 µM Spt6$^{1247-1451}$. Crystals were grown at 4°C by vapor diffusion. Drops comprised 0.4 µl of 14 mg/mL protein and 0.2 µl well solution (40% ethanol, 100 mM Tris-Cl pH 7.0; #7 of the Emerald BioSystems Cryo II screen; Rigaku, Bainbridge Island, WA). Crystals were cooled by plunging into liquid nitrogen. For the Spt6$^{1247-1451}$-Rpb1$^{1468-1500\ pT1471,pS1493}$ complex, 47 µM Spt6$^{1247-1451}$ was mixed with a 5-fold molar excess of crude Rpb1$^{1468-1500\ pT1471,pS1493}$ and concentrated to 500 µM Spt6$^{1247-1451}$. Crystals were grown at 4°C by vapor diffusion. Drops comprised 0.4 µl of 14 mg/mL protein and 0.2 µl well solution (10% isopropanol, 0.1 M Na HEPES pH 7.5, 20% PEG 4000; #41 of the Crystal Screen HT; Hampton Research, Aliso Viejo, CA). For both structures, data were collected on a Rigaku MicroMax-007HF copper rotating-anode generator with VariMax-HR confocal optic, using a Rigaku R-AXIS IV ++ detector. Data were integrated and scaled with HKL2000 (RRID:SCR_015547) (*Otwinowski and Minor, 1997*). The structures were determined by molecular replacement with Phaser (CCP4 program suite; RRID:SCR_014219) (*McCoy et al., 2007*) using the coordinates of unbound Spt6 tSH2 domain (PDB: 3PSJ) as the search model. Model building and refinement were performed in COOT (RRID:SCR_014222) (*Emsley et al., 2010*) and Phenix (RRID:SCR_014224) (*Adams et al., 2010*), respectively. Figures of molecular structures were prepared using UCSF Chimera (RRID:SCR_004097) (*Pettersen et al., 2004*).

## Quantitative ChIP-seq (qChIP-seq)

qChIP-seq was performed essentially as described previously (*Ramakrishnan et al., 2016*). Six independent cultures of the wild type control and three independent cultures of the *spt6-R-H*$^{(T/Y)}$, *spt6-KK-AA*$^{(S)}$, *spt6-R-H*$^{(T/Y)}$,*KK-AA*$^{(S)}$, *rpb1-FSP*$^+$*-KKR*$^+$, *rpb1-Y-A,FSP*$^+$*KKR* + , and *rpb1-T-A,FSP*$^+$*-KKR*$^+$- strains were grown to mid-log phase. Cells were subjected to cross-linking with formaldehyde (1%) for 20 min at room temperature. Glycine (0.125 mM) was added to quench crosslinking and cells were washed with 1X phosphate-buffered saline. For the reference genome, *Candida glabrata* (ATCC 36909; American Type Culture Collection, Manassas, VA) was grown at 25°C and cross-linked with formaldehyde as described above. Cross-linked *S. cerevisiae* (40 × 10$^7$) and *C. glabrata* (20 × 10$^7$) cells were mixed in FA140 +SDS buffer (50 mM HEPES-KOH, pH 7.5, 140 mM NaCl, 1 mM EDTA, 1% Triton X-100, 0.1% sodium deoxycholate, 0.1% SDS) and lysed by bead beating. Lysates from two sets of *S. cerevisiae* and *C. glabrata* mixture (120 × 10$^7$ cells) were pooled for sonication using a Diagenode Bioruptor (6 cycles of 5 min each, 30 s ON and 30 s OFF at high setting). Soluble chromatin was obtained by two sequential high-speed centrifugations. To remove SDS, soluble chromatin obtained from four batches of yeast cells were pooled and transferred to a Spectra-Por 5 mL Float-A-Lyzer G2 dialysis device (3.5–5 kD MWCO; Spectrum Labs, Rancho Dominguez, CA), dialyzed initially for 2 hr in 1 L FA buffer (50 mM HEPES-KOH, pH 7.5, 1 mM EDTA, 1% Triton X-100, 0.1% sodium deoxycholate) with 140 mM NaCl followed by an overnight dialysis against 4 L of the same buffer. The dialysate was centrifuged at 16,000 RCF for 15 min at 4°C. A 100 µl aliquot was set aside for isolating input DNA. The remaining dialysate was split equally into three tubes and pre-cleared using 50 µl of blocked Dynabeads Protein A for 1 hr at 4°C with end-over-end rotation. Blocking of Dynabeads Protein A beads was performed at 4°C with end-over-end rotation for 4 hr using FA buffer containing 140 mM NaCl, bovine serum albumin (BSA) (0.2 mg/mL; New England Biolabs, Ipswich, MA) and fish skin gelatin (0.5%, Sigma-Alrich). Pre-cleared dialysate was combined with 30 µL α-Spt6 antibody (*McCullough et al., 2015*; RRID:AB_2687439). Cross-reactivity of the antibody, which was raised against *S. cerevisiae* Spt6, for *C. glabrata* Spt6 was verified by western

blot (*Figure 6—figure supplement 2*). For a background control, the additional set of pre-cleared dialysate prepared from the wild type control strain was combined with 30 µL pre-immune serum. 100 µl blocked Dynabeads Protein A was then added and incubated overnight at 4°C with end-over-end rotation. Beads were sequentially washed 5 min each with end-over-end rotation at 4°C with FA buffer +140 mM NaCl (three washes), FA buffer +1000 mM NaCl (two washes), FA buffer +500 mM NaCl (two washes), and one wash each with LiCl/NP40 buffer (10 mM Tris-Cl, pH 8.0, 250 mM LiCl, 0.5% NP40, 0.5% sodium deoxycholate) and TE (10 mM Tris-Cl, pH 8.0, 1 mM EDTA). Elution and reverse crosslinking of ChIP and input DNA followed by RNase A and Proteinase K digestion steps were performed as described previously (*Chandrasekharan et al., 2011*). ChIP and input DNA were purified using the Qiagen MinElute PCR Purification kit according to the manufacturer's instructions, except two washes each with buffer QG and PE were performed. Sequencing libraries were constructed using NEBNext Multiplex Oligos (Index Primers Set 1 and Set 2) and NEBNext ChIP-seq Library Prep Reagent Set for Illumina (New England Biolabs), and subjected to 50 bp single-end sequencing in an Illumina Hiseq2000 sequencer (Illumina, San Diego, CA).

## Bioinformatics analysis

Illumina sequencing reads for each biological replicate were sequentially aligned to both the *S. cerevisiae* genome (SGD release R64, UCSC SacCer3) and the *C. glabrata* genome (Ensembl Fungi release 30, GCA_000002545.2) using Novoalign (RRID:SCR_014818), allowing for one random assignment of multi-mapping reads. Coordinate duplicate alignments were retained to ensure an accurate occupancy profile for the abundant Spt6 protein after careful inspection for potential PCR artifacts and consideration of unpublished results. Species-specific and orthologous alignments were extracted by comparing the alignment metrics for every read to either genome with a custom program, cross_species_alignment_picker (*Parnell, 2016*); orthologous alignments were discarded for analysis. Fragment coverage for *S. cerevisiae* was generated with BioToolBox bam2wig with an extension of 130 bp, a consensus fragment length as determined by peak cross-strand correlation, and depth-normalized by scaling to reads per million mapped (RPM). These coverage tracks were further normalized for ChIP efficiency by dividing by the number of unique *C. glabrata* alignments (scaled to millions) in each sample (*Orlando et al., 2014*). After normalization, a mean consensus track was generated by merging the replicates using bedtools (RRID:SCR_006646) and dividing by the number of replicates. To generate fold enrichment tracks, genomic intervals with zero coverage in the input samples were artificially set to a non-zero value of 1 (a value well below the genomic mean) and a fold enrichment calculated with macs2 bdgcmp. Coverage track manipulation was performed using the script manipulate_wig.

For high-level gene analysis from all ChIP experiments, fold enrichment was mapped across gene bodies in 40 bins (representing 2.5% of gene length) plus 25 10 bp bins upstream and downstream using the BioToolBox program get_binned_data. Subsequent analysis and data processing steps were performed using programs available within the BioToolBox package. Graphs were generated with GraphPad Prism (GraphPad Software, Inc., La Jolla, CA; RRID:SCR_002798). Gene transcription rates were derived from *Pelechano et al, (2010)*.

## Data, reagent, and software availability

The qChIP-seq data have been deposited at the NCBI Gene Expression Omnibus (*Edgar et al., 2002*) under accession number GSE98405 (https://www.ncbi.nlm.nih.gov/geo/query/acc.cgi?acc=GSE98405).

The accession numbers for the atomic coordinates and structure factors of the tSH2-Rpb1 linker structures are PDB: 5VKL and 5VKO.

The custom program cross_species_alignment_picker is available at GitHub (https://github.com/tjparnell/HCI-Scripts/blob/master/BamFile/cross_species_alignment_picker.pl). A copy is archived at https://github.com/elifesciences-publications/HCI-Scripts.

Plasmids have been deposited at Addgene (see *Supplementary file 1*).

## Acknowledgements

We thank Zaily Connell and Laura McCullough for technical assistance with genetic studies, and the following individuals and funding associated with University of Utah Core Facilities: Peptide

Synthesis, Scott Endicott; Mass Spectrometry, Krishna Parsawar, (S10RR020883 and S10RR025532); Bioinformatics, Tim Parnell, (P30CA042014). This project was supported by R01GM116560 to TF and CPH and P50GM082545 to CPH.

## Additional information

### Competing interests
Tim Formosa: Reviewing editor, *eLife*. The other authors declare that no competing interests exist.

### Funding

| Funder | Grant reference number | Author |
|---|---|---|
| National Institutes of Health | R01GM116560 | Matthew Allan Sdano<br>Frank G Whitby<br>Tim Formosa<br>Christopher P Hill |
| National Institutes of Health | P50GM082545 | Matthew Allan Sdano<br>James M Fulcher<br>Frank G Whitby<br>Christopher P Hill |
| National Institutes of Health | P30CA042014 | Timothy J Parnell |

The funders had no role in study design, data collection and interpretation, or the decision to submit the work for publication.

### Author contributions
Matthew A Sdano, Conceptualization, Formal analysis, Validation, Investigation, Visualization, Writing—original draft, Writing—review and editing; James M Fulcher, Resources, Writing—review and editing; Sowmiya Palani, Mahesh B Chandrasekharan, Investigation, Methodology, Writing—review and editing; Timothy J Parnell, Data curation, Formal analysis, Validation, Writing—review and editing; Frank G Whitby, Formal analysis, Validation, Investigation; Tim Formosa, Conceptualization, Supervision, Funding acquisition, Validation, Investigation, Visualization, Writing—original draft, Project administration, Writing—review and editing; Christopher P Hill, Conceptualization, Supervision, Funding acquisition, Writing—original draft, Project administration, Writing—review and editing

### Author ORCIDs
Matthew A Sdano http://orcid.org/0000-0001-6702-2755
James M Fulcher http://orcid.org/0000-0001-9033-3623
Tim Formosa http://orcid.org/0000-0002-8477-2483
Christopher P Hill https://orcid.org/0000-0001-6796-7740

### Decision letter and Author response
Decision letter https://doi.org/10.7554/eLife.28723.033
Author response https://doi.org/10.7554/eLife.28723.034

## Additional files

### Supplementary files
• Supplementary file 1. Plasmids.
DOI: https://doi.org/10.7554/eLife.28723.023

• Supplementary file 2. Yeast strains.
DOI: https://doi.org/10.7554/eLife.28723.024

• Transparent reporting form
DOI: https://doi.org/10.7554/eLife.28723.025

## Major datasets

The following datasets were generated:

| Author(s) | Year | Dataset title | Dataset URL | Database, license, and accessibility information |
|---|---|---|---|---|
| Sdano MA, Fulcher JM, Palani S, Chandrasekharan MB, Whitby FG, Formosa T, Hill CP | 2017 | Genome-wide maps of Spt6 occupancy in yeast strains with mutations in Spt6 and Rpb1 | https://www.ncbi.nlm.nih.gov/geo/query/acc.cgi?acc=GSE98405 | Publicly available at the NCBI Gene Expression Omnibus (accession no: GSE98405) |
| Sdano MA, Whitby FG, Hill CP | 2017 | SPT6 tSH2-RPB1 1476-1500 pS1493 | http://www.rcsb.org/pdb/explore/explore.do?structureId=5vkl | Publicly available at the RCSB Protein Data Bank (accession no: 5VKL) |
| Sdano MA, Whitby FG, Hill CP | 2017 | SPT6 tSH2-RPB1 1468-1500 pT1471, pS1493 | http://www.rcsb.org/pdb/explore/explore.do?structureId=5vko | Publicly available at the RCSB Protein Data Bank (accession no: 5VKO) |

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
