## [Decision Letter]

Thank you for submitting your article "A novel SH2 recognition mechanism recruits Spt6 to the doubly phosphorylated Rpb1 linker at sites of transcription" for consideration by *eLife*. Your article has been very favorably reviewed by three peer reviewers, and the evaluation has been overseen by John Kuriyan as the Reviewing Editor. The following individuals involved in review of your submission have agreed to reveal their identity: Bruce J Mayer (Reviewer #1); Michael J Eck (Reviewer #2).

The reviewers have discussed the reviews with one another and the Reviewing Editor has drafted this decision to help you prepare a revised submission.

This manuscript is an excellent combination of in vitro biochemistry, structural biology, and in vivo biology that reveals details of the interaction between the SH2 domains of *S. cerevisiae* Spt6 with the RNA PolII subunit Rpb1. Spt6 contains the only known SH2 domains in yeast, arranged in a tandem module, which are thought to represent the progenitor domain from which the many phosphotyrosine-binding SH2 domains present in metazoans evolved. The authors find that this module binds to a singly or doubly phosphorylated segment in the linker region of Rbp1, rather than to the heptad repeat region of the CTD, contrary to the prevailing model.

An X-ray crystal structure of the tandem SH2 domains with phosphorylated Rpb1 linker peptide showed that the primary binding surface to the phosphopeptide is not the canonical phosphotyrosine binding pocket of the N-terminal SH2 domain, but instead a novel peptide binding groove on the C-terminal cryptic SH2 domain (which lacks many of the hallmark residues common to other SH2 domains). They go on to show that longer Rpb1 peptides, containing additional N-terminal phosphorylation sites, bind with higher affinity than the minimal binding peptide; a crystal structure reveals that the N-terminal phosphosite interacts with the canonical pTyr binding pocket of the N-terminal SH2. Furthermore, the phosphate of phosphothreonine and the ring of a nearby tyrosine residue together bind to the pTyr pocket of the SH2 domain in a fashion that closely mimics binding of phosphotyrosine in metazoan SH2 domains. This raises an intriguing possibility that this primordial binding activity, optimized for binding the Rpb1 linker, may have been co-opted for new signaling purposes once dedicated tyrosine kinases evolved. Finally, as expected, mutations in either Spt6 or Rpb1 designed to impair the two binding interfaces impaired known Spt6-mediated phenotypes in yeast.

The study is very well done and will be important addition to the field, as it overturns accepted wisdom and provides structural evidence for a new paradigm for SH2 domain recognition of Y..pS instead of pY ligands. The structural work is first rate, and is complemented by careful binding studies with phosphopeptides (singly and doubly phosphorylated) derived from the newly discovered binding site. Mass spectrometry studies confirm phosphorylation of these sites in vivo, and the functional relevance of the Spt6 interaction at the site is demonstrated via extensive structure-function work and ChIP-seq experiments that show the interaction is important for maintaining repressive chromatin.

Overall, this is a thorough and convincing work, that provides important new understanding of the repertoire of SH2 domain interactions. The work is eminently suitable for publication in *eLife*.

Please consider the following points raised by the reviewers in preparing a revised manuscript. Note that although some of the points that are raised might require new experimentation to address the issues completely, we are not requiring further experimentation as a condition for acceptance of the manuscript. You should use your judgement as to how to best address the issues.

Important points to address:

1) The absence of any binding data for extended peptides that are phosphorylated on the N-terminal site (T1471 or Y1473) but not S1493 hampers the ability to understand the extent to which the "canonical" site cooperates with the S1493 binding site. In particular, it would be very helpful to know whether phosphorylation of the N-terminal site is sufficient to mediate some specific binding even in the absence of S1493 phosphorylation. This would help interpret the in vivo data (Figure 5). The reviewers do not think that this is absolutely essential, but feel that it would make the story more complete, and help in interpreting the in vivo binding data. If such data are not readily available, please provide an explanation along with the revised manuscript.

2) The authors should discuss the measured binding affinities in the context of the high concentration of Spt6 present in vivo (as noted in the Discussion). By a back-of-the-envelope calculation, the nuclear concentration of Spt6 is ~10 μM, high above the dissociation constant for singly phosphorylated (S1493) peptide. One might expect binding would be saturated even in the absence of the phosphorylation of the second Rpb1 site.

3) Subsection “The Spt6-Rpb1 interaction is important for maintaining repressive chromatin”: The fact that mutations in Rpb1 and Spt6 affecting the same binding interface have additive effects is difficult to understand-if the effect if the Spt6-KK-AA mutant is failure to coordinate the phosphate of pS1493, addition of the Rpb1S1493A mutation in the double mutant should have no additional phenotype.

4) Subsection “The Rpb1 linker recruits Spt6 to highly transcribed transcription units”: "One likely explanation for this observation is that these less severe mutations allow relatively normal recruitment of Spt6 to transcription sites, but cause a defect in RNAPII progression, which would result in increased RNAPII (and therefore Spt6) retention/occupancy."

There does not appear to be specific evidence for "likely"- this should be tested or moved to discussion as speculation. It might be important to consider Rpb1 or Rpb3 or other Pol II subunit controls for the claim that recruitment of Spt6 is defective in the interface mutants- under that model there is unstated assumption that Pol II levels are not changing when occupancy of Spt6 is reduced, though altered Pol II levels are invoked to explain increased Spt6 occupancy in weak interface mutants.

---

## [Author Response]

Important points to address:1) The absence of any binding data for extended peptides that are phosphorylated on the N-terminal site (T1471 or Y1473) but not S1493 hampers the ability to understand the extent to which the "canonical" site cooperates with the S1493 binding site. In particular, it would be very helpful to know whether phosphorylation of the N-terminal site is sufficient to mediate some specific binding even in the absence of S1493 phosphorylation. This would help interpret the in vivo data (Figure 5). The reviewers do not think that this is absolutely essential, but feel that it would make the story more complete, and help in interpreting the in vivo binding data. If such data are not readily available, please provide an explanation along with the revised manuscript.

We have added the following sentences to the Results subsections of the revised manuscript: 1) “Phosphorylation of Rpb1 T1471 or Y1473 enhance Spt6 binding” and 2) “The Spt6-Rpb1 interaction is important for maintaining repressive chromatin”:

1) “The singly phosphorylated pY1473 peptide (Fl-Rpb1^1468-1500 pY1473^) did not bind the tSH2 domain (up to 10 μM tSH2; data not shown), indicating that pY1473 is less important for binding than pS1493. Because pT1471 contributes similar binding energy as pY1473 in the doubly phosphorylated peptides, we do not expect the singly phosphorylated pT1471 peptide to bind either, although we have not tested this directly.”

2) “The appearance of phenotypes upon mutating the pS interface but not the pT/pY interface is consistent with the greater contribution of pS1493 to binding(Figure 4).”

2) The authors should discuss the measured binding affinities in the context of the high concentration of Spt6 present in vivo (as noted in the Discussion). By a back-of-the-envelope calculation, the nuclear concentration of Spt6 is ~10 μM, high above the dissociation constant for singly phosphorylated (S1493) peptide. One might expect binding would be saturated even in the absence of the phosphorylation of the second Rpb1 site.

We agree that this is an interesting point that merits more explicit discussion. The revised manuscript includes the following segment in the Discussion section:

“As noted above, Spt6 is highly abundant in the yeast nucleus (~10 μM (McCullough et al., 2015)), which raises interesting questions about the affinities we observed for biochemical binding of phosphorylated peptides. In particular, the 100 nM affinity of the singly phosphorylated pS1943 peptide might be sufficient to ensure saturation of Spt6-Rpb1 binding in the absence of a reinforcing interaction from phosphorylation at T1471 or Y1473. One possibility is that the Spt6 tSH2 domain has additional ligands in the cell, and that T1471 or Y1473 phosphorylation is required to drive Rpb1 association in the face of competing interactions.”

3) Subsection “The Spt6-Rpb1 interaction is important for maintaining repressive chromatin”: The fact that mutations in Rpb1 and Spt6 affecting the same binding interface have additive effects is difficult to understand-if the effect if the Spt6-KK-AA mutant is failure to coordinate the phosphate of pS1493, addition of the Rpb1S1493A mutation in the double mutant should have no additional phenotype.

We agree, and have added the following text in the revised manuscript (Figure 5—figure supplement 1 legend):

“The additive effects of these alleles suggest that these residues may have additional functions outside of this specific interaction.”

4) Subsection “The Rpb1 linker recruits Spt6 to highly transcribed transcription units”: "One likely explanation for this observation is that these less severe mutations allow relatively normal recruitment of Spt6 to transcription sites, but cause a defect in RNAPII progression, which would result in increased RNAPII (and therefore Spt6) retention/occupancy."There does not appear to be specific evidence for "likely"- this should be tested or moved to discussion as speculation. It might be important to consider Rpb1 or Rpb3 or other Pol II subunit controls for the claim that recruitment of Spt6 is defective in the interface mutants- under that model there is unstated assumption that Pol II levels are not changing when occupancy of Spt6 is reduced, though altered Pol II levels are invoked to explain increased Spt6 occupancy in weak interface mutants.

We agree and have moved the discussion of this phenomenon to the Discussion section, which now reads:

“In contrast to the severe Spt6 and Rpb1 mutants, less severe mutations that impair a single phosphate-binding site increase Spt6 occupancy across gene bodies. One possible explanation for this observation is that these less severe mutations allow relatively normal recruitment of Spt6 to transcription sites, but cause a defect in RNAPII progression that results in increased RNAPII (and therefore Spt6) retention/occupancy. Our data also include the intriguing observation that Spt6 occupancy is skewed toward the 3' end of average genes, and that the accumulation of Spt6 directly over the transcription termination site is diminished or lost in our binding interface mutants (Figure 6), which may reflect a role for this interaction in termination of transcription. Definitive explanations for these various correlations will require further study.”